# *Withania somnifera* Ameliorates Doxorubicin-Induced Nephrotoxicity and Potentiates Its Therapeutic Efficacy Targeting SIRT1/Nrf2, Oxidative Stress, Inflammation, and Apoptosis

**DOI:** 10.3390/ph18020248

**Published:** 2025-02-12

**Authors:** Amany Mohammed Mohmmed Hegab, Soha Osama Hassanin, Reham Hassan Mekky, Samah Sulaiman Abuzahrah, Alaaeldin Ahmed Hamza, Iman M. Talaat, Amr Amin

**Affiliations:** 1Developmental Pharmacology and Acute Toxicity Department, Egyptian Drug Authority (EDA), Formerly National Organization of Drug Control and Research (NODCAR), Giza 12611, Egypt; manalhegab2630@gmail.com; 2Biochemistry Department, Faculty of Pharmacy, Modern University for Technology and Information, Cairo 11585, Egypt; soha.hassanin@pharm.mti.edu.eg; 3Department of Pharmacognosy, Faculty of Pharmacy, Egyptian Russian University, Badr City, Cairo-Suez Road, Cairo 11829, Egypt; reham-mekky@eru.edu.eg; 4Samah Sulaiman Abuzahrah, Department of Biological Sciences, College of Science, University of Jeddah, Jeddah 21959, Saudi Arabia; sabozahera@uj.edu.sa; 5Biology Department, Egyptian Drug Authority (EDA), Formerly National Organization of Drug Control and Research (NODCAR), Giza 12611, Egypt; 6Clinical Sciences Department, College of Medicine, University of Sharjah, Sharjah 27272, United Arab Emirates; italaat@sharjah.ac.ae; 7Research Institute for Medical and Health Sciences, University of Sharjah, Sharjah 27272, United Arab Emirates; 8Pathology Department, Alexandria University, Alexandria 22113, Egypt; 9Department of Basic Medical Sciences, College of Medicine, University of Sharjah, Sharjah 27272, United Arab Emirates

**Keywords:** doxorubicin, *Withania somnifera*, nephrotoxicity, anti-inflammatory, therapeutic efficacy, oxidative stress

## Abstract

**Background**: Doxorubicin (DOX) is a very powerful chemotherapy drug. However, its severe toxicity and potential for resistance development limit its application. *Withania somnifera* L. Dunal (WIT) has therapeutic capacities, including anti-inflammatory, antioxidant, and anticancer activities. This study investigates the preventative benefits of a standardized WIT extract against DOX-induced renal damage in vivo. We also investigate the synergistic effects of combining WIT and DOX to improve therapeutic efficacy in breast cancer cells (MCF7-ADR). **Methods:** This study employed an animal model where rats were administered 300 mg/kg/day of WIT orally for a duration of 14 days. Rats received DOX injections at a dose of 5 mg/kg, for a total of 15 mg, on the 6th, 8th, and 10th days. **Results**: Present results revealed that WIT reduced DOX-induced increase levels of blood urea and creatinine and the activity of kidney injury molecule-1. WIT also reduced renal tissue damage, oxidative stress, and levels of pro-inflammatory markers. WIT alleviated the effects of DOX on nuclear factor erythroid 2-related factor 2, heme oxygenase-1, and sirtuin 1 in the renal tissues. WIT modulated nuclear factor-κB activity and decreased apoptotic indicators. Furthermore, WIT improves DOX’s capacity to kill drug-resistant MCF7-ADR cells by arresting the cell cycle and promoting apoptosis. Chemical analysis of WIT root extract revealed 34 distinct compounds, including alkaloids, withanolides, flavanones, and fatty acids. **Conclusions**: These constituents synergistically contribute to WIT’s antioxidant, anti-inflammatory, and anti-apoptotic properties. In addition, they confirm its ability to reduce systemic toxicity while improving treatment efficacy.

## 1. Introduction

Doxorubicin (DOX), an anthracycline antibiotic, is a potent anti-cancer drug used to treat various tumor types [1]. The non-specific toxicity and drug-induced desensitization emerging from resistance in different cancer cells have primarily limited the use of DOX therapy [1,2,3]. Among its non-specific toxicity, DOX is known to adversely affect other vital organs, including the kidneys [3,4,5,6]. DOX’s damaging effect on the kidneys is mediated by generating free radicals, causing oxidative damage, and triggering apoptosis and inflammation. Pro-inflammatory cytokines and inflammatory mediators, including NF-kB, TNF-α, and interleukin-1β, have a role in the initiation of DOX-induced nephrotoxicity [7,8]. It is, therefore, recommended to utilize additional agents alongside DOX therapy to minimize the harm inflicted on tissues by its sole clinical application [3].

Research has shown that natural ingredients can reduce inadvertent harm to essential tissues while improving the efficacy of anti-cancer drugs [3,9,10,11]. *Withania somnifera* L. Dunal (WIT), commonly referred to as Indian ginseng or ashwagandha, is a member of the Solanaceae family. Traditional and Ayurvedic medicine has extensively employed it throughout its long history [12,13]. Notable WIT applications include its ability to enhance libido, regulate the immune system, induce relaxation, improve liver function, combat aging, alleviate depression, and reduce inflammation [12,14]. Moreover, WIT has many therapeutic properties, such as anti-arthritic, hypoglycemic, hepatoprotective, neuroprotective, and anti-cancer effects [13,15].

WIT can alleviate oxidative stress and inflammation caused by different chemicals in several animal models. It effectively prevents hepatic encephalopathy caused by thioacetamide [16], bronchial asthma induced by ovalbumin [17], neurotoxicity induced by methylenedioxymethamphetamine [18], urotoxicity caused by cyclophosphamide [19], cardiotoxicity caused by DOX [20], hepatotoxicity caused by bromobenzene [21], and renal toxicity caused by gentamicin and bromobenzene [22,23]. Breast cancer cells exhibited reduced growth in nude mice when administered 300 mg/kg of WIT root extract. In addition, it stopped breast cancer cells from multiplying and stopped the production of cytokines that help cancer spread in MDA-MB-231 cells [24]. Singh et al. [25] and Tewari et al. [15] conducted studies demonstrating the therapeutic efficacy of phytochemicals derived from WIT against various types of cancer. Withanolides (WA), a potent bioactive component found in the leaves and root extracts of WIT, offers a wide range of biological benefits in different animal models of breast cancer. These effects include anti-inflammatory, antioxidant, and anti-carcinogenic capabilities [26]. Multiple studies have demonstrated that combining WIT and WA improves the effectiveness and responsiveness of anti-neoplastic drugs in breast cancer cells, such as cisplatin and paclitaxel, leading to enhanced anti-tumor activity [27,28,29]. This combined effect is often mediated by inducing apoptosis and inhibiting the proliferation of cancer cells. Furthermore, preliminary research suggests that WIT has the potential to enhance the immune system’s capacity to combat cancer, promote cellular adaptability, and facilitate cellular regeneration, making it a viable adjunctive therapy for cancer [12,29]. However, limited research has explored the potential synergistic effect of combining WIT with DOX in targeting breast cancer cells.

This study evaluated the potential renal-protective efficacy of WIT against DOX-induced nephrotoxicity in an established animal model. We further examined whether the co-administration of DOX and WIT could enhance cytotoxicity against breast cancer cells in vitro and exert a synergistic therapeutic effect.

## 2. Results

### 2.1. Metabolic Profiling of WIT Root Extract by LC-MS and Tandem MS/MS

The chemical composition of the WIT root extract was analyzed using RP-HPLC-MS and tandem MS/MS in the positive ionization mode. This analysis showed 34 different metabolites. Figure 1a illustrates the base peak chromatogram (BPC) of the extract; the metabolites were characterized based on retention times (RT), generated molecular formulae, molecular ion peaks (*m*/*z*), neutral losses, double bond equivalence (DBE), and areas of the peaks (Figure 1b) [30,31]. Relevant literature and databases [32,33,34,35,36] were consulted during the characterization process. The identified metabolites were categorized as alkaloids (10), withanolides (16), fatty acids (7), and flavanone (Figure 1b,c, Table 1). As for alkaloids, they mainly belonged to tropane alkaloids, with the fragment of *m*/*z* 124.11 assigned to the tropane nucleus, a common feature in the alkaloids of Withania and Solanaceae in general. The observed alkaloids were calystegin A3, pelletierine/isopelletierine, N-methylpelletierine, cuscohygrine, tigloidine isomers I–II, 3-methylbutyryloxytropane isomers I–III, and withasomnine. It bears noting that alkaloids were the most abundant class, with 53.4% of the relative abundance (Table 1).

Regarding withanolides, 16 derivatives were detected in the extract of the roots of Withania somnifera, a C28 steroidal lactone, where the lactone rings were six-membered, with a C–C bond to the molecule (Figure 1c) [37]. They exhibited the sequential loss of water moieties, accompanied by hexosyl moieties in the case of glycosides. Their molecular ions were observed with sodium adducts [M+H+Na]+ (Table 1). In this line, withaferin A and withanolide A were the major withanolides in terms of relative abundance, where their fragmentation and elution, as well as their constitutional isomers 12-deoxywithastramonolide and withanone, are in agreement with previously published studies [34,35]. Also, withanolide L and 2,3-dihydrowithaferin A were characterized in WIT (Table 1).

Regarding glycosides, two isomers of physagulin D were noticed as an example of withanolides conjugated to a single sugar moiety. Meanwhile, withanoside V, withanoside IV/VI isomers I–IV, and withanone II are examples of conjugation to two sugar moieties (Table 1, Figure 1c). In addition, two isomers of ashwagandhanolide (C56H78O12S) were detected, a dimeric withanolide with a characteristic thioether link [38]. Concerning fatty acids, palmitic acid and octadecanoic acid were noticed alongside the unsaturated derivatives of the latter, *viz*., oleic acid, linoleic acid, and linolenic acid isomers I–III [33,39]. Moreover, farnesylnaringenin was annotated in WIT for the first time, where it was described before in Solanaceae [36].

### 2.2. Effect of DOX and WIT on Body and Kidney Weights and Relative Kidney Weights

Administration of DOX led to significant decreases in final body weight, body gain, kidney weight, and relative kidney weight compared to the normal control group (Table 2). In contrast, when WIT was administered alongside DOX to rats, there was a substantial improvement in final body weight, kidney weight, and relative kidney weight compared to the groups treated with DOX. However, there was a notable reduction in body weight gain.

### 2.3. WIT Enhances Renal Function and Reduces Levels of the KIM-1 Injury Marker

Albumin, urea, and creatinine levels in the serum were assessed to monitor changes in kidney function, whereas renal KIM-1 was used as an indicator of kidney damage (Figure 2). The administration of DOX led to elevated serum creatinine and urea levels, together with increased kidney KIM-1 concentrations, relative to the control group (Figure 2A–C). In addition, the treatment of DOX led to a significant reduction in serum albumin concentration relative to the control group, as illustrated in Figure 2D. Nevertheless, the simultaneous administration of WIT successfully alleviated the rise in urea, creatinine, and KIM-1 concentrations and reversed the decline in albumin levels compared to the DOX group. The administration of WIT alone at a dosage of 300 mg/kg body weight showed levels comparable to the control, suggesting that the plant extract does not cause toxicity.

### 2.4. WIT Decreases the Histological Changes in the Renal Tissue of Rats Treated with DOX

Photomicrographs of kidney sections from rats in the control and WIT groups demonstrate a typical architectural pattern with undamaged glomeruli (G) and intact proximal (P) and distal (D) tubules (Figure 3 and Table 3). Compared to control rats, kidneys of DOX-treated rats showed notable damage to the glomeruli and tubules. Among the kidney problems that have been described are moderate glomerular capsule shrinkage and distortion, tubular dilatation with protein cast deposition, severe vacuolization, and tubular degeneration. Significant blood vessel congestion with perivascular edema and fibrosis was observed in the cortical interstice, along with focal hemorrhages and inflammatory cell infiltrations between degenerative tubules. However, sections from rats treated with DOX+WIT exhibit almost normal glomerular and tubular architecture, with very little glomerular capsule atrophy, tubular dilatation, and degenerations, and the presence of protein casts (score 0.6–1).

### 2.5. WIT Reduces Apoptosis and the Presence of Apoptotic Markers in the Kidneys of Rats Treated with DOX

The TUNEL assay was employed to detect apoptotic cells in renal tissue, with brown staining signifying TUNEL-positive nuclei. The control and WIT-treated rats showed minimal apoptosis, as shown in Figure 4A,B. However, there was a clear rise in the number of TUNEL-positive cells in the DOX group compared to the control group (Figure 4A,B). The glomeruli and tubules of the DOX-treated cohort housed the majority of TUNEL-positive cells. Fewer TUNEL-positive cells were found in the group that was given WIT and DOX together than in the DOX-treated group. In both the control and WIT groups, the amounts of BAX and caspase-3 proteins in the kidney homogenate were about the same. The DOX group showed a significant increase in the apoptotic marker BAX (Figure 4C) and caspase-3 (Figure 4D) compared to the control. Conversely, administration of WIT and DOX led to a notable decrease in the protein levels of BAX and caspase-3 compared to the DOX group.

### 2.6. WIT Reduces Oxidative Stress Levels in the Kidneys of Rats Treated with DOX

Rats intoxicated with DOX had a notable elevation in renal MDA and P. Carbonyl levels and MPO activity, along with a considerable reduction in GSH concentration, TAC, and the activities of SOD and CAT enzymes relative to the normal control group (Figure 5). Nevertheless, simultaneous administration of WIT and DOX led to a significant reduction in renal tissue levels of MDA and P. Carbonyl, as well as a reduction in MPO activity. In addition, it elevated the concentration of renal tissue GSH and TAC, and the activity of SOD and CAT enzymes, in comparison to the control and DOX groups. No significant alterations were reported between the WIT and control groups for the levels of MDA, P. Carbonyl, MPO, and non-enzymatic GSH in renal tissue. Additionally, there were no significant changes in the TAC or the activities of the antioxidant enzymes SOD and CAT. The rats that received DOX alone had a substantial elevation of calcium in their kidneys, as seen in Figure 5H. Nevertheless, the calcium concentration in the kidneys did not show a substantial deviation from normal values when WIT was administered before DOX. However, the administration of WIT alone did not cause any significant change in renal calcium levels compared to the control.

### 2.7. WIT Attenuates Inflammatory Markers in the Kidneys of DOX-Treated Rats

Inflammatory markers, including NF-kB, TNF-α, IL-1ß, and IL-6, were evaluated to determine the anti-inflammatory effects of WIT on rats treated with DOX. Rats administered DOX demonstrated a significant elevation in renal NF-kB (Figure 6A), TNF-α (Figure 6B), IL-1β (Figure 6C), and IL-6 (Figure 6D) levels relative to the normal control group. Nonetheless, the concurrent administration of WIT with DOX markedly reduced renal concentrations of NF-kB, TNF-α, IL-1β, and IL-6 compared to rats treated only with DOX.

### 2.8. WIT Upregulates Renal Expressions of Nrf2, Ho-1, and SIRT1 in Rats Treated with DOX

The RT-PCR analysis revealed a significant reduction in renal Nrf-2, HO-1, and SIRT1 mRNA transcripts in the DOX group relative to the control group (Figure 7A–C). Nevertheless, there was a significant increase in Nrf-2, HO-1, and SIRT1 mRNA transcripts in the WIT and DOX groups. Specifically, these levels escalated by 3.3-, 2.3-, and 3.8-fold, respectively, compared to the group solely administered DOX (Figure 7). In addition, ELISA was used to evaluate the effect of WIT on the renal protein expression of Nrf-2 (Figure 7D) and SIRT1 (Figure 7E). The administration of DOX caused a significant decrease in the expression of both Nrf2 and SIRT1 in the kidneys, as seen in Figure 7A,B, compared to the control group. In contrast, administering WIT and DOX significantly enhanced the expressions of renal Nrf-2 (Figure 7A) and SIRT1 (Figure 7B) compared to the rats intoxicated with DOX. Significantly, the levels of both Nrf2 and SIRT1 were unaffected by treatment with WIT alone, compared to the control group (Figure 7).

### 2.9. Different Viability of MCF-7, MCF-7/ADR, and MCF-10A Cells with WIT and DOX

The initial investigation demonstrated the cytotoxic effects of DOX and WIT on MCF-7 and drug-resistant MCF-7/ADR cells. The viability of MCF-7, MCF-7/ADR, and MCF-10A cells was assessed using the MTT assay following a 24 h treatment with varying concentrations of DOX and WIT (Figure 8A–C).

The findings indicated that DOX and WIT displayed cytotoxic characteristics against every cell line examined, resulting in a dose-dependent reduction in cell viability. Both MCF-7 and MCF-7/ADR cells showed substantial cytotoxicity after receiving DOX treatment, and both sensitive and resistant cells showed considerable growth suppression. The DOX IC_50_ values were 4.50 ± 0.07 μg/mL in MCF-7 cells, 13.36 ± 0.56 μg/mL in MCF-7/ADR cells, and 14.26 ± 2.10 μg/mL in MCF-10A cells (Figure 8B). With IC_50_ values of 96.73 ± 2.65 μg/mL and 152.80 ± 3.40 μg/mL, respectively, WIT similarly suppressed the growth of MCF-7 and MCF-7/ADR cells. On MCF-10A cells, however, its toxicity was lower (IC_50_ = 238.5 ± 3.48 μg/mL Figure 8C,D). According to these findings, DOX showed substantial toxicity in both malignant and normal cells, whereas WIT showed moderate cytotoxicity on MCF-7 and MCF-7/ADR cells and low toxicity on normal cells. Additionally, chemo-sensitive MCF-7 cells had IC_50_ values of 1.58 for WIT and nearly 3-fold for DOX, while drug-resistant MCF-7/ADR cells had IC_50_ values that were noticeably greater. MCF-7/ADR cells were chosen for additional research based on these results, as well as the IC_50_ values of WIT and DOX.

To explore the potential synergistic effect of WIT with DOX on cell growth inhibition, the combination index (CI) was determined at varying concentrations of DOX combined with 10 and 20 µg/mL of WIT. CompuSyn software was employed for this analysis, revealing a moderate synergistic effect between WIT and DOX (CI = 0.81~0.62), with the lowest CI observed at 10 and 20 µg/mL of WIT combined with 3 and 4 µg/mL of DOX (Figure 8E). Therefore, these concentrations of WIT were used in the subsequent MTT assays. Interestingly, combining WIT at 10 and 20 µg/mL before challenging with DOX increased the growth inhibitory effects of DOX in the MCF-7/ADR cell line. In the DOX and WIT combinations, DOX at a dose of 6.25 µg/mL decreased the proliferation of MCF-7/ADR cells from 20% to 68 and 90%, respectively (Figure 8G). Combining DOX with WIT (10 and 20 µg/mL) reduced the IC_50_ value from 13.36 µg/mL to 1.49 and 1.05 µg/mL (Figure 8F). Additionally, the proliferation data of the different DOX concentrations, combined with WIT at 10 and 20 µg/mL, were evaluated by CompuSyn software to search for the possible synergistic effect of WIT on DOX-suppressed cell growth. We found a synergistic pattern across all DOX/WIT concentration combinations, with IC values less than 0.5 (Figure 8H). The results showed that WIT synergistically reduces cell proliferation and reverses DOX-induced resistance in the MCF-7/ADR cells, resulting in increased sensitivity to DOX exposure.

### 2.10. WIT and DOX Combination Increases Apoptosis and G2/M Cell Cycle Arrest in MCF-7/ADR Cells

The study looked at how WIT (20 μg/mL), DOX (6.2 μg/mL), or a combination of WIT (10 and 20 μg/mL) and DOX (6.2 μg/mL) affected apoptosis and necrosis in MCF-7/ADR cells over 24 h. Prior to analysis using flow cytometry, we stained the cells with propidium iodide (PI) and Annexin V (Figure 9A). The flow cytometry study of the control group of MCF-7/ADR cells demonstrated 0.93% apoptosis and 2.28% necrosis. Using WIT caused a considerable rise in apoptosis, reaching 15.58%. Apoptosis rates increased in the early and late stages (12.05% and 3.53%, respectively). DOX treatment significantly increased late apoptosis by 13.91% (*p* < 0.05), total apoptosis by 20.37% (*p* < 0.05), and necrosis by 6.21% compared to WIT-treated cells. Adding 10 μg/mL of WIT to DOX-treated cells significantly increased (*p* < 0.05) late apoptosis (15.95%), total apoptosis (21.55%), and necrosis (9.17%) compared to DOX alone. Administration of a solution containing 20 μg/mL of WIT and DOX resulted in significant increases (*p* < 0.05) in the percentages of early apoptosis (8.85%), late apoptosis (19.51%), total apoptosis (28.35%), and necrotic cells (8.12%) (Figure 9A,C). Administering DOX and WIT at 20 μg/mL significantly increased total cell death from 26.91% to 36.37%.

Flow cytometry was employed to investigate cell cycle progression in MCF-7/ADR cells subjected to varying doses of WIT and DOX, administered either individually or in conjunction. The objective was to ascertain if the inhibitory effects of DOX and WIT on cellular proliferation were associated with cell cycle arrest (Figure 9B,D). Following a 24 h WIT treatment, the percentage of tumor cells in the G2/M phase significantly rose from 8.81% to 21.23% in comparison to control cells. DOX treatment significantly increased the percentage of tumor cells in the G2/M phase, rising from 8.81% to 35.17%. In conjunction with WIT, DOX at 10 μg/mL significantly elevated the quantity of tumor cells in the G2/M phase. In comparison to WIT treatment alone, the percentage significantly increased within 24 h, from 21.23% to 28.66%. It took 24 h for 20 μg/mL of WIT to raise the number of tumor cells in the G2/M phase from 35.17% to 43.33% compared to DOX alone. The results show that treating MCF-7/ADR cells with DOX and WIT effectively stops them from moving forward in the G2/M phase of the cell cycle, as shown in Figure 9B–D.

## 3. Discussion

DOX, a potent chemotherapeutic drug, has been used to treat numerous cancer types [1]. The risk of toxicity and resistance is a significant downside of its application [1,2]. Healthcare specialists are often concerned about the severe effects that DOX may have on essential organs, notably the kidneys and heart [1,2,3]. As a result, there is an urgent need to produce adjuvant drugs that minimize side effects while increasing treatment efficacy [40,41]. The present findings show that WIT has significant therapeutic potential for reducing DOX-induced nephrotoxicity. We studied the effect of WIT on an acute renal damage model. In our animal models, a decrease in body weight, kidney weight, and relative kidney weight served as indications of acute renal injury. Concomitant with these alterations were impaired renal function, elevated levels of a kidney damage biomarker, diminished antioxidant activity, and heightened indicators of renal oxidative stress and proinflammatory markers. The histological study indicated extensive glomerular and tubular damage. We observed that the WIT extract had a high potential for decreasing DOX-induced cytotoxicity in rat kidneys. According to our findings, WIT reduced renal oxidative stress while boosting the kidney’s ability to protect against free radicals. These reduce inflammation and apoptosis, protecting the kidneys from DOX-induced injury.

DOX-induced renal damage is characterized by severe lesions in the glomeruli, proximal and distal convoluted tubules, a decrease in glomerular filtration rate (GFR), high serum urea and creatinine levels, and hypoalbuminemia [4,5,6]. In the current study, acute renal damage was reported in rats administered DOX. Rafiee et al. [5] identified high blood creatinine and urea levels as renal failure and glomerular injury indicators. DOX-treated rats exhibit reduced glomerular filtration, resulting in elevated serum urea and creatinine levels. Rats may develop nephrotic syndrome, defined by a significant alteration in the glomerular filtration barrier. Because of this disruption, many proteins may be eliminated in the urine, resulting in hypoalbuminemia and hypoproteinemia [42]. Previous studies [43,44] have shown that DOX-nephrotoxic rats had an elevated renal concentration of KIM-1, indicating kidney injury. KIM-1 is a transmembrane protein found in damaged parts of proximal renal tubular epithelial cells. It was found to be an early sign of kidney damage [45]. In the current study, DOX administration significantly increases KIM-1 expression. DOX-induced kidney histopathology revealed pathological abnormalities such as necrobiosis, renal blood vessel congestion, and vacuolization. According to our findings, WIT showed substantial nephroprotective effect against renal damage and dysfunction caused by DOX. It minimized the DOX-induced reduction in kidney weight and reverse biochemical and histological abnormalities. These findings support previous research demonstrating the effectiveness of natural chemicals in protecting rats from kidney injury, and provide more evidence of WIT’s nephroprotective properties [11]. The present findings support our initial notion, which has been reinforced by previous research indicating that WIT may preserve kidney tubule and glomerulus structures while lowering renal damage indicators in various animal models [22,23,46].

DOX’s anthracycline ring structure causes oxidative stress, a crucial pathophysiological mechanism in DOX-induced kidney damage [1]. Gille and Hans [47] discovered that DOX facilitates both enzymatic and nonenzymatic single-electron redox cycles, resulting in the release of reactive oxygen species (ROS) from molecular oxygen. Oxidative stress occurs parallel with the increased ROS generation caused by DOX exposure. Depletion of GSH, an important antioxidant, reduced levels of antioxidant enzyme activities, and higher contents of MDA, a lipid peroxidation marker, P. Carbonyl, a protein oxidation marker, and MPO activity, a sign of inflammation and oxidative stress, are just a few examples of how oxidative stress manifests itself [48]. Previous investigations have validated similar findings [4,7,43]. The current study identified DOX-induced oxidative stress by measuring an increase in MDA, P. Carbonyl, and MPO. Furthermore, DOX reduced the kidney’s TAC and GSH levels, as well as the antioxidant capability of the CAT and SOD enzymes. Nonetheless, combined treatment of DOX and WIT restored antioxidant levels and markers of renal oxidative stress to normal. These results align with new research that shows WIT extracts can lower oxidative stress caused by several drugs, such as thioacetamide, gentamicin, and bromobenzene [16,21,46]. Tetali et al. [14] showed that the main active ingredient in WIT is steroidal lactones, also known as withanolides. These are what give the treatment its antioxidant effect.

ROS triggers apoptosis as it damages mitochondria and activates sensitive signaling pathways and their ability to degrade cellular macromolecules directly [5,49]. Apoptosis is reported as a significant factor in DOX-induced nephrotoxicity [7,50,51]. Data presented here supported these previous reports and additionally suggested that WIT’s anti-apoptotic and antioxidant properties may be closely related. Indeed, it has been revealed that Withaferin A (WFA), derived from WIT, inhibits H_2_O_2_-induced apoptosis in cardiomyocytes [52]. Both necrotic and apoptotic cell death can be influenced by elevated intracellular calcium levels. Ca^2+^ influxes into the cytoplasm from internal cellular reserves and extracellular areas as a result of ROS damaging calcium channels [53,54]. Calcium excess is hypothesized to cause necrotic cell death by activating cellular proteases, nucleases, and lipases [54]. Additionally, the accumulation of intracellular calcium is linked to caspase activation and the subsequent accumulation of mitochondrial calcium, leading to apoptotic cell death [54,55]. In this work, DOX raised intracellular calcium levels in the renal tissues of rats, whereas WIT treatment led to a marked reduction in tissue calcium content. This protective effect may be ascribed to the antioxidant properties or the calcium-antagonistic characteristics identified in prior studies [56].

Renal tissue accumulation of inflammatory cells is another method responsible for creating ROS. This study discovered that rats administered DOX showed increased renal MPO activity, indicating acute inflammation and leukocyte buildup in the tissues. The research indicated that DOX elevated the concentrations of inflammatory mediators, including NF-kB, IL-1β, TNF-α, and IL-6. This is consistent with previous studies [4,5,8,57] that showed DOX’s function in encouraging the overproduction of superoxide and ROS, which activates the NF-kB pathway and aids in releasing proinflammatory cytokines. The study found that WIT effectively reduced renal MPO activity and NF-kB, IL-1β, TNF-α, and IL-6 levels in DOX-induced kidney damage. This means that WIT may be able to protect against DOX-induced nephrotoxicity because it can lower inflammation and neutrophil infiltration. Previous in vitro and animal investigations have demonstrated that WIT root has anti-inflammatory and immunosuppressive characteristics [58,59,60].

Nrf2 is a crucial transcription factor that regulates the expression of various antioxidant genes, including SOD, CAT, and HO-1, in reaction to oxidative stress [61,62]. Furthermore, when stimulated, Nrf2 may move from the cytosol to the nucleus, increasing the activity of antioxidant enzymes and decreasing ROS-induced damage [61,63]. DOX showed a downregulatory impact on Nrf2 and HO-1 expression in rat kidneys. This is consistent with previous data suggesting that DOX downregulates the Nrf2/HO-1 pathway [4,64,65,66]. However, WIT counteracted DOX’s downregulatory impact, enhancing Nrf2 and HO-1 expression in renal tissues. The protective function of Nrf2 is further supported by the positive effects of natural compounds known to activate it, such as polyphenols, terpenoids, and alkaloids [67]. The increase in antioxidant activity seen with WIT may have something to do with the functionalization of the Nrf2/HO-1 signaling pathway in the kidneys of rats that were administered DOX. SIRT1, an NAD+-dependent deacetylase, exhibits decreased expression and activity during inflammation [68] and in acute kidney injury induced by various agents [4,69,70,71]. To mechanistically confirm the protective effects of WIT, we assessed SIRT1 in the DOX administration. We found considerably reduced levels of SIRT1 in the DOX group compared to the control, which is consistent with earlier findings [71,72]. A potent nephroprotective SIRT1 is highly expressed in renal tubules. It deacetylates many transcription factors by lowering oxidative damage, inflammation, fibrosis, and apoptosis in renal disorders [73,74,75]. Previous research has shown that increasing SIRT1 expression restores cellular oxidant-antioxidant equilibrium and reduces kidney damage in mouse models of acute renal injury [70,74,76,77]. SIRT1 reduces inflammation and boosts antioxidants by inhibiting NF-κB and activating Nrf2 [73,78]. In this investigation, WIT increased SIRT1 gene and protein expression while increasing Nrf2 and HO-1 expression. Notably, SIRT1 increases the activity of the Nrf2/antioxidant response element (ARE) pathway. NF-κB is deacetylated and turned off when SIRT1 expression goes up, and that ceases the inflammatory response. Upregulation of SIRT1 protein expression modifies NF-κB signaling in acute renal damage caused by cadmium [77], DOX [72], and diclofenac [79] in rats. We conclude that SIRT1 suppression, which appears to be a significant therapeutic target for WIT, mediates DOX’s oxidative, inflammatory, and apoptotic effects.

The WIT extract’s antioxidant, anti-inflammatory, and anti-apoptotic properties are most likely due to the presence of key bioactive compounds found during HPLC analysis. These metabolites can act alone or in combination to provide therapeutic effects. Various phytochemical investigations have shown the presence of over 12 alkaloids, 40 withanolides, and various steroids in different parts of the WIT plant [13,34,80]. In this study, 34 metabolites were identified and further investigated in WIT root extract. The principal chemicals in WIT extracts, such as withanoside V, withanoside IV, 12-deoxywithastramonolide, withanolide A, and withaferin A, play essential roles in a variety of biological activities, including antioxidative and anticancer properties [25,81]. Due to their unusual steroidal structure and potent bioactivities, withanolides are particularly appealing as primary chemicals for treating inflammatory diseases and cancer [82]. In the end, DOX raised MPA activity, intracellular calcium, and end products of lipid and protein oxidation while lowering TAC, GSH, and SOD levels in the kidney tissue. Histopathological results in this group, including kidney necrosis and apoptosis, validated the impact of nephrotoxicity. WIT therapy prevented animals from developing nephrotoxicity due to DOX and reduced oxidative damage. WIT’s preventive impact might be ascribed to its antioxidant, calcium-antagonistic, and anti-inflammatory properties (Figure 10).

DOX treatment can cause nonspecific toxicity and resistance, limiting its efficacy in therapy [1,2,3]. Employing effective medicine combinations to prevent chemoresistance in cancer treatment is crucial. Combination therapy, which targets many pathways, can improve treatment efficacy and reduce the systemic toxicity of chemotherapeutic medications [3,10,11]. The secondary objective of this investigation was to ascertain if WIT may enhance the anticancer efficacy of DOX in breast cancer cells, encompassing both drug-sensitive and drug-resistant cell lines (MCF-7 and MCF-7/MDR). We investigated the DOX-WIT combination’s capacity to suppress cell growth and determined inhibitory concentrations using the MTT method. In our MTT analysis, we discovered that WIT extract is moderately hazardous to MCF-7 cells. In addition, DOX and WIT treatments have less efficacy on MCF-7/ADR cells. We focused on the potential of WIT to surmount drug resistance in a DOX-resistant breast cancer cell line, specifically MCF7- ADR cells. The results of the MTT assay suggest that DOX, over a wide range of concentrations, has a synergistic antiproliferative effect on MCF7-ADR cells when applied in combination with WIT at small concentrations (10 and 20 μg/mL). As a result, we chose a non-toxic dose of WIT in the following studies to better assess its effect on MCF7-ADR’s chemosensitivity to DOX. The present study discovered that DOX and WIT have a synergistic anticancer effect.

Flow cytometry was employed to detect apoptosis and cell cycle progression, allowing us to thoroughly study how WIT and DOX inhibition affects MCF7-ADR cell growth. Using WIT during therapy significantly improved DOX-induced cell growth inhibition. Additionally, administering DOX and WIT to MCF7-ADR cells significantly increased apoptotic rates. The results clearly demonstrate that WIT can improve DOX’s ability to trigger apoptosis in MCF7-ADR cells. A recent study has indicated that the utilization of WIT hydroalcoholic extracts in conjunction with paclitaxel (PTX) yields superior outcomes in terms of sensitivity and efficacy compared to PTX alone. This was observed in both in vitro breast cancer cells and animal models with breast cancer cell grafts [29]. Furthermore, researchers evaluated many combinations of withaferin A with conventional chemotherapy drugs, such as DOX, oxaliplatin, and cisplatin, in various cancer types [83]. A low dose of DOX combined with a low dose of withaferin A may help treat ovarian cancer due to its synergistic effects. The synergistic effect is due to withaferin A’s capacity to increase DOX sensitivity, resulting in cell death via ROS-induced autophagy. Fong et al. [83] discovered that this technique requires the activation of both LC3B and caspase-3. According to studies, withaferin A enhances oxaliplatin’s capacity to suppress cell growth and induce cell death in human pancreatic cancer cells. According to Li et al. [84] withaferin A suppresses the PI3K/AKT pathway, which causes mitochondrial dysfunction. This is the source of the observed rise. Furthermore, Hahm et al. [27] showed that using withaferin A makes breast cancer cells more susceptible to the effects of cisplatin on cell proliferation and death. The findings suggest the use of withaferin A as an additional therapy to enhance the efficacy of standard chemotherapy drugs in a number of cancer types. Furthermore, we demonstrated that WIT alone has the ability to impact the regulation of the cell cycle in breast cells, in addition to its role in inducing apoptosis and promoting cell death. We observed that DOX and WIT significantly induced a G2/M-phase arrest compared to DOX or WIT alone. The findings indicate that WIT can modify the distribution of the tumor cell cycle and enhance apoptosis. So, WIT might be a good addition or chemosensitizer to make DOX more effective at killing cancer cells in the MCF7-ADR model. The encouraging outcomes of WIT with active constituents and their efficacy in enhancing DOX’s effectiveness against cancer cells necessitate implementing a novel drug development strategy known as “Proteolysis Targeting Chimera” (PROTAC) [85]. Currently, reported PROTACs for target discovery predominantly emphasize natural products and pharmaceuticals. Using PROTAC technology in drug target discovery represents a significant research avenue, particularly in identifying targets for natural products and novel drugs. A new multi-target identification approach termed “Degradation-Based Protein Profiling” (DBPP) may facilitate the identification of targets for these diverse, active compounds in WIT and develop promising agents for cancer treatment, inflammation targeting, and other beneficial outcomes.

## 4. Materials and Methods

### 4.1. Chemicals

Ashwagandha root extract powder from WIT was acquired from Solgar Inc. (500 Willow Tree Road, Leonia, NJ, USA, batch number SOLGB78064 03D). Methanol, acetonitrile, and glacial acetic acid were procured from Fisher Chemicals (ThermoFisher, Waltham, MA, USA). The solvents employed for characterization were of HPLC-MS grade, respectively. Ultrapure water was procured utilizing a Milli-Q system (Millipore, Bedford, MA, USA). The reagent kits for serum albumin (ALB, 1001023), creatinine (1001110), and urea (41040) were acquired from Bio-Diagnostic (Cairo, Egypt). The Annexin V-FITC Apoptosis Detection Kit (Bio Vision Research Products, 980 Linda Vista Avenue, Mountain View, CA, USA). Propidium iodide flow cytometry kit for cell cycle analysis (Abcam, ab139418, Cambridge, UK). Fetal bovine serum (FBS), streptomycin, Dulbecco’s Modified Eagle Medium, 2-[4-(2-hydroxyethyl) piperazin-1-yl] ethane sulfonic acid (HEPES), glutamine, and penicillin were sourced from Gibco/Invitrogen (Karlsruhe, Germany). The 3-(4,5-dimethylthiazol-2-yl)-2,5-diphenyltetrazolium bromide (MTT) utilized in the MTT test kit for in vitro toxicity was acquired from Sigma (Sigma, St. Louis, MO, USA, M-5655). Doxorubicin hydrochloride was acquired from Hikma Specialized Pharmaceuticals, Egypt. The Apoptag Plus Peroxidase In Situ Apoptosis Detection Kit is made by Chemicon International, Temecula, CA, USA. We used ELISA kits for tumor necrosis factor-α (TNF-α, My BioSource—MBS355371, San Diego, CA, USA), interleukin-1 beta (IL-1β, My BioSource—MBS825017), interleukin-6 (IL-6, My BioSource—MBS355410), nuclear factor-κB (NF-κB, My BioSource—MBS453975), nuclear factor erythroid 2-related factor 2 (Nrf2, My BioSource—MBS012148), and Sirtuin-1 (SIRT1, My BioSource—MBS775316). The QIAamp RNeasy Mini kit for total RNA extraction is made by Qiagen GmbH, Hilden, Germany. The additional compounds were acquired from Sigma Chemical Co. in St. Louis, MO, USA: 2,4-dinitrophenylhydrazine, thiobarbituric acid (TBA), Folin’s reagent, epinephrine, 5,5-dithiobis (2-nitrobenzoic acid), superoxide dismutase (SOD) enzyme, hydrogen peroxide (H_2_O_2_), and bovine serum albumin. The final chemicals were acquired from local commercial providers.

### 4.2. Analysis WIT Characterization Using RP-HPLC-ESI-QTOF-MS and -MS/MS

The WIT extract was reconstituted in 80% MeOH/water before subjecting it to the RP-HPLC analysis. The analysis was conducted using an Agilent 1200 series quick resolution system (Agilent Technologies, Santa Clara, CA, USA), which consisted of a quaternary pump (G7104C) and an autosampler (G7129A). The separation was conducted using a Poroshell 120 HiLiC Plus column (150 mm × 3 mm, 2.7 μm particle size, Agilent Technologies). As previously described, the system was connected to a 6530-quadruple time of flight (Q-TOF) LC/MS (Agilent Technologies) with a dual ESI interface. The mobile phases utilized for gradient elution comprised acetonitrile (designated as phase B) and acidified water (containing 0.5% acetic acid, *v*/*v*), administered at a flow rate of 0.2 mL/min. The replicates of the extracts were subjected to analysis, utilizing an injection volume of 5 µL. The process of gradient elimination was conducted as follows: At 0 min, the composition was 99% A and 1% B; at 5.50 min, it was 93% A and 7% B; at 11 min, the ratio was 86% A and 14% B; at 17.50 min, it was 76% A and 24% B; at 22.50 min, the distribution was 60% A and 40% B; at 27.50 min, the composition shifted to 0% A and 100% B; at 28.50 min, it remained at 0% A and 100% B; at 29.50 min, the ratio returned to 99% A and 1% B; and finally, at 40 min, it was again 99% A and 1% B. The subsequent operating conditions were succinctly outlined: a drying nitrogen gas temperature of 325 °C with a flow rate of 10 L/min; a nebulizer pressure of 20 psig; a sheath gas temperature of 400 °C with a flow rate of 12 L/min; a capillary voltage of 4000 V; a nozzle voltage of 500 V; a fragmentor voltage of 130 V; a skimmer voltage of 45 V; and an octapole radiofrequency voltage of 750 V. The MassHunter Workstation software B.06.00, developed by Agilent Technologies, was employed to manage data acquisition in profile mode at a frequency of 2.5 Hz. Spectra were obtained in positive ionization mode across a mass-to-charge (*m*/*z*) range of 70 to 1100. The detection threshold was set at 100 parts per million (ppm) [86]. MassHunter Qualitative Analysis B.06.00 (Agilent Technologies) software was employed for data analysis and characterization [87,88]. For the generated molecular formulae, retention times (RT), molecular ion peaks (*m*/*z*), and fragmentation patterns were observed. The Reaxys database (https://www.reaxys.com, accessed on 30 November 2024), and KNApSAcK Core System database (http://www.knapsackfamily.com/knapsack_core/top.php, accessed on 30 November 2024), were consulted to cross-reference the data that was obtained. Furthermore, pertinent literature was procured from the Egyptian Knowledge Bank (https://www.ekb.eg, accessed on 30 November 2024). Statistical analysis was performed using Microsoft Excel 365 (Redmond, WA, USA) and Minitab 17 (Minitab, Inc., State College, PA, USA).

### 4.3. Animals

Adult male Wistar albino rats weighing 175 to 200 g were obtained from the National Organization for Drug Control and Research (NODCAR) animal facility in Giza, Egypt. The rats were allowed to acclimate to their environment for a week before to the start of the experiments. They were provided unrestricted water access and a standardized diet of rat pellets. The rats were carefully kept in polycarbonate enclosures with wood chip bedding, subjected to a 12 h light/dark cycle, and kept in a temperature-controlled chamber at a consistent temperature of 22 to 24 °C. The research followed the criteria established by the US National Institutes of Health for the appropriate care and use of laboratory animals. The animal ethics committee approved the animal care and handling protocol at NODCAR (Ethics approval number: NODCAR/1 January 2024).

### 4.4. Drugs

Doxorubicin hydrochloride (DOX) was acquired as a vial called Adricin, with a dosage of 50 mg from Hikma Specialized Pharmaceuticals, Cairo, Egypt. The Ashwagandha (*Withania somnifera*) root powder, with the batch number SOLGB78064 03D, was obtained from Solgar Inc. at 500 Willow Tree Road, Leonia, NJ, USA. The ashwagandha pill’s herbal recipe consists of 300 mg of WIT root extract, usually 1.5% withanolides, equivalent to 4.5 mg of each capsule.

### 4.5. Determining the Appropriate Dosage

DOX was used to develop an animal model for kidney damage, and it was mixed with physiological saline just before animal administration. The rats received a dosage of 5 mg/kg through three equal injections at 48 h intervals. This resulted in a cumulative dose of 15 mg/kg b.wt. for each rat. This dosage has been extensively studied and proven to induce nephrotoxicity [51,89,90]. The rats orally received the WIT pill via gavage at a dose of 300 mg/kg b.wt. or 4.5 mg of withanolides, which was suspended in distilled water. The administration volume was 5 mL/ kg b.wt. The dose and method of administration were chosen based on previous research showing that WIT extract can protect the kidneys from damage caused by gentamicin and bromobenzene [23,46], as well as its hepatoprotective properties against oxidative stress in rats induced by thioacetamide and bromobenzene [16,21]. The quantity and schedule employed in our prior investigation were validated by the research done by Hamza et al. [20] and Jeyanthi et al. [22].

### 4.6. Treatment Protocol

The rats received the following treatments after being divided into four groups of six at random:Control group: The rats received 5 mL/kg b.wt. of distilled water for 14 days. After a 5-day interval, they received intraperitoneal (i.p.) injections of 5 mL/kg saline on the 6th, 8th, and 10th days.WIT group: The rats were administered an oral dose of 300 mg/kg body weight of WIT extract for 14 days. After a 5-day interval, they received i.p. injections of 5 mL/kg saline on the 6th, 8th, and 10th days.DOX treatment group: The animals were given distilled water by mouth for 14 days, and then had three i.p. injections of 5 mg/kg DOX on the 6th, 8th, and 10th days, resulting in a total dosage of 15 mg/kg.The DOX + WIT treatment group: The rats were administered an oral dose of 300 mg/kg body weight of WIT extract for 14 days. Three i.p. injections of DOX (5 mg/kg) were administered on the 6th, 8th, and 10th days, resulting in a cumulative dosage of 15 mg/kg.

Renal tissue and blood samples were obtained from each group 24 h after the final injections of WIT, DOX, and the control solution for further examination.

### 4.7. Preparation of the Sample

The rats received 3% sodium pentobarbital (45 mg/kg, i.p.) to induce anesthesia; this was done 24 h after the treatment of WIT and four days after the injection of DOX. Later, blood specimens were collected from the retro-orbital plexus. After blood extraction, the rats were euthanized via cervical dislocation. The kidneys were immediately removed, washed with cold normal saline, and weighed. The relative tissue weight is calculated by multiplying the tissue weight by 100 and dividing it by the body weight. The left kidney was immediately placed in a solution of 10% buffered formalin to aid in further histological studies. In order to simplify the process of biochemical testing, the right kidney was crushed into a uniform mixture using a Tris-HCL buffer. The ratio used was 1 part kidney tissue to 10 parts buffer, with the buffer having a concentration of 150 mM and a pH of 7.4. Throughout the process, a low temperature was maintained. Subsequently to homogenization, the mixture was aliquoted into vials and stored at a low temperature of −20 °C. The levels of reduced glutathione (GSH), malondialdehyde (MDA), protein carbonyls (P. carbonyl), total proteins, myeloperoxidase (MPO), superoxide dismutase (SOD), catalase (CAT) activity, inflammatory indicators, and calcium were measured by diluting the samples in the indicated buffers. Blood samples were collected using centrifuge tubes and then centrifuged at a speed of 3000 revolutions per minute for 20 min in a chilled centrifuge set at a temperature of 4 °C. This procedure was conducted to obtain the serum.

### 4.8. Assessment of Kidney Function

We employed the PerkinElmer Lambda 25 UV/VIS spectrophotometer, PerkinElmer Inc., 940 Winter Street Waltham, MA, USA to measure the concentrations of serum albumin (ALB, 1001023), creatinine (1001110), and urea (41040) to assess renal function. The tests were performed using bio-diagnostic reagent kits from Cairo, Egypt. The steps specified in the manual accompanying the kit, as given by the manufacturer, were adequately followed.

### 4.9. Evaluation of Kidney Injury Molecule-1 (KIM-1)

The concentration of KIM-1 in the kidney homogenate was quantified utilizing enzyme-linked immunosorbent assay (ELISA) technology, adhering to the manufacturer’s guidelines from Nordic BioSites AB-Sweden, Nordic Biosite Inc., Wayne, PA, USA. The results were measured in picograms per milligram (in pg/mg) of tissue.

### 4.10. Evaluation of Oxidative Stress in the Kidneys

Different approaches were utilized to identify indications of renal oxidative stress. The Van Doorn et al. technique [91] determined the glutathione concentration (GSH) in the kidney homogenate. This method relies on the interaction between the thiol group of glutathione (GSH) at pH 8.0 and Ellman’s reagent, 5,5-dithiobis (2-nitrobenzoic acid). This interaction leads to the creation of a yellow hue, which indicates the existence of the 5-thiol-2-nitrobenzoate anion. The quantification of MDA, a biomarker for lipid peroxidation, was performed following the approach outlined by Uchiyama and Mihara in 1978 [92]. This technique was based on creating a pink substance with maximum absorption at 535 nm by the reaction between MDA and thiobarbituric acid (TBA). The Aebi approach [93] was employed to evaluate catalase (CAT) activity. This approach involves monitoring the fast reduction of H_2_O_2_ at a designated wavelength of 240 nm and reporting the results in units per mg of protein. The process for assessing the activity of the SOD enzyme in the kidney homogenate was developed by Sun and Zigman [94] by using the enzyme’s ability to impede the spontaneous oxidation of adrenaline at high pH levels.

The method employed by Reznick and Packer [95] involved measuring the amount of P. carbonyl by producing protein hydrazones at a wavelength of 370 nm through the reaction between 2,4-dinitrophenylhydrazine and protein carbonyls. The quantification of the data was expressed as nm of carbonyl groups/mg protein, using a molar extinction value of 22,000 M/cm. The kidney’s total antioxidant capacity (TAC) was measured using the ferric reducing antioxidant power (FRAP) assay. The test conducted by Benzie and Strain [96] uses the measurement of absorbance at 593 nm to detect the creation of a blue-colored complex of ferrous-tripyridyl triazine. The presence of antioxidants in the sample facilitates the development of this complex by providing electrons to aid in the reduction of the colorless oxidized ferric form.

Kidney myeloperoxidase activity was quantified using a methodology previously elucidated by Hillegass et al. [97]. In summary, 2.9 mL of potassium phosphate buffer with a concentration of 50 mmol/L and a pH of 6.0 was combined with 0.167 mg/mL of lo-dianisidine dihydrochloride and 0.0005% hydrogen peroxide. Subsequently, this combination was mixed with 100 μL of supernatant. The change in light absorption at a wavelength of 460 nm was measured using a spectrophotometer. A unit of MPO was defined as the decomposition of one micromole of peroxide per minute.

### 4.11. Evaluation of Calcium and Total Protein Levels in the Kidney

The calcium content in the kidney was measured using a Varian Vista-MPXCCD Simultaneous ICP-OES EVISA, Göttingen, Germany, which employs i nductively coupled plasma optical emission spectrometry (ICP-OES). The kidney samples were fully digested by combining distilled nitric acid and perchloric acid, gradually raising the temperature to 200 °C until the process was finished.

The Lowry technique, as adapted by Peterson [98], was employed to assess the total protein content of the kidneys. A PerkinElmer Lambda 25 UV/VIS spectrophotometer was employed in each test to measure absorbance.

### 4.12. Histopathological Analysis

The kidney tissue was conserved by submerging it in a 10% neutral formaldehyde solution for fixing. The sample was then cut into slices with a thickness of 5 μm using a microtome (Leica RM2255, Tokyo, Japan) and stored in paraffin for preservation. The sections were histologically assessed using hematoxylin and eosin (H&E). We utilized a Leica DMRB/E light microscope to examine the sections. The histologist evaluated the extent of kidney damage using the criteria specified by Altınkaynak et al. [50]. These criteria include the presence of interstitial inflammatory infiltration, congestion in blood vessels, perivascular edema, fibrosis, localized glomerular atrophy, dilatation of the Bowman capsule, tubular dilatation, and accumulation of intratubular casts, degeneration of tubular epithelial cells, and necrosis of tubular epithelial cells. The ultimate evaluation included a 0–3 scale to quantify the magnitude of the damage, with higher scores indicating more significant harm.

The score was determined using the subsequent scale:

Score 0 = No damage.

Score 1 = Localized damage, less than 25%; slight modifications.

Score 2 = Damage ranging from 25% to 50%, with the observed changes occurring in many locations and being of moderate intensity.

Score 3 = Damage surpasses 50%; severe modifications.

### 4.13. Evaluation of Apoptotic Cells in the Kidney

The Apoptag plus Peroxidase in Situ Apoptosis Detection Kit from Chemicon International (Temecula, CA, USA) was utilized in accordance with the manufacturer’s guidelines to assess apoptosis in deparaffinized sections. The TUNEL method used terminal deoxynucleotidyl transferase to label the ends of 3-OH DNA with digoxigenin nucleotides. This method facilitates the identification of DNA fragmentation associated with apoptosis. Afterward, the fragments that have been tagged attach to peroxidase-conjugated anti-digoxigenin antibodies. The specimens exhibited color development upon the application of a peroxidase substrate. The quantification of apoptotic cells in each slice was conducted by enumerating the number of TUNEL-positive apoptotic cells in 10 specific areas per slide, utilizing a 400× magnification.

### 4.14. Assessment of Indicators of Apoptosis

The levels of Bcl-2–associated X protein, BAX (0.078–20 ng/mL), and Caspase-3 (0.068–20 ng/mL) in the kidneys were measured using rat-specific ELISA kits: Bax (LS Bio-LS-F21494) and Caspase-3 (LS Bio-LS-F4138), respectively. The experiments were performed according to the directions supplied by Lifespan Biosciences, Inc. (2 Shaker Rd Suites, Shirley, MA, USA). The protein content in the tissue homogenate was quantified using the Bradford technique following the established protocol.

### 4.15. Measurement of Inflammatory Biomarkers

The levels of inflammation markers in kidney tissue were evaluated using ELISA kits, specifically, tumor necrosis factor-α (TNF-α, My BioSource—MBS355371), interleukin-1 beta (IL-1β, My BioSource—MBS825017), interleukin-6 (IL-6, My BioSource—MBS355410), and NF-KB (My BioSource—MBS453975). The concentrations of TNF-α and IL-6 ranged from 1.9 to 250 pg/mL, whereas NF-KB levels were detected between 0.62 and 80 ng/mL. The studies followed the methods provided by the manufacturer Lifespan Biosciences, Inc. (2 Shaker Rd Suites, Shirley, MA, USA). The experiments were completed carefully using a microplate reader, namely, the Spectra Max i3X from Molecular Devices in San Jose, CA, USA. The manufacturer’s instructions were followed precisely. Each measurement was conducted three times to guarantee precision and dependability. In addition, the protein content in the tissue homogenate was assessed using the Bradford technique, following the established protocol.

### 4.16. Analysis of Gene Expression for Nrf2, HO-1, and Sirt-1

Renal Nrf2, HO-1, and Sirtuin-1 (Sirt-1) gene expression study was performed using RT-PCR. The QIAamp RNeasy Mini kit, manufactured by Qiagen GmbH in Germany, was used to extract total RNA from 30 mg of kidney tissues, following the instructions provided by the manufacturer. The assessment of the purification of the extracted RNA specimens was conducted using the QIAamp RNeasy Mini kit (Qiagen, Germany, GmbH). The primers were acquired from Metabion (Planegg, Germany) and were chosen based on sequences accessible on the website http://www.ncbi.nlm.nih.gov/tools/primer-blast, accessed on 30 November 2024. The forward primer sequence for β-actin is 5′–TCCTCCTGAGCGCAAGTACTCT-3′, while the reverse primer sequence is 5′–CTCTGCTCAGTAACAGTCCGCCTAGAA-3′. The forward primer sequence for Nrf2 is 5′–CACATCCAGACAGACACCAGT-3′, while the reverse primer sequence is 5′–CTACAAATGGGAATGTCTCTGC-3. The forward primer for HO-1 is 5′–GGCTTTAAGCTGGTGATGGC-3′, while the reverse primer sequence is 5′– GGGTTCTGCTTGTTTCGCTC -3′. The forward sequence for SIRT1 is 5′–CAC-CAG-AAA-GAA-CTT-CAC-CAC-CAG-3′, and the reverse sequence is 5′–ACC-ATC-AAG-CCG-CCT-ACT-AAT-CTG-3. A 96-well RT-PCR plate was utilized for a reaction volume of 25 µL. The reaction mixture comprised 12.5 µL of 2× QuantiTect SYBR Green PCR Master Mix (Qiagen, Germany, GmbH), 0.25 µL of RevertAid Reverse Transcriptase (200 U/µL) (ThermoFisher), 0.5 µL of each primer at a concentration of 20 pmol, 8.25 µL of water, and 3 µL of RNA template. The experiment utilized a Stratagene MX3005P real-time PCR equipment produced by Agilent Technologies in Sydney, Australia. The amplification curves and CT values were determined using Stratagene MX3005P software version 1.6.0. The thermal cycling regimen included 30 min at 50 °C for cDNA synthesis and 15 min at 94 °C for polymerase activation and reverse transcriptase inactivation. The plates experienced 40 amplification cycles under these conditions: an initial denaturation at 94 °C for 15 s, followed by annealing at 94 °C for 15 s and 72 °C for 30 s. A melting curve investigation was thereafter conducted by incrementally increasing the temperature from 60 °C to 95 °C. The expression levels of each target gene were normalized by comparing them to the average cycle threshold values of the housekeeping gene Actin. To evaluate the disparities in gene expression among different groups, the CT value of each sample was compared to that of the normal control group utilizing the “ΔΔCt” approach, as described by Yuan et al. [99].

### 4.17. Evaluate the Protein Concentrations of Nrf2 and Sirt-1 in the Renal Tissue

Analysis was conducted on the protein concentrations of Nrf2 and Sirt-1 in renal homogenates. The levels of Nrf-2 and SIRT1 were measured using rat ELISA kits (#MBS012148 and #MBS775316; MyBioSource). The quantification was done using the procedures supplied by the manufacturer, Lifespan Biosciences, Inc. (2 Shaker Rd Suites, Shirley, MA, USA). The experiments were conducted with great attention to detail using a microplate reader Spectra Max i3X from Molecular Devices, San Jose, CA, USA, following the manufacturer’s instructions precisely. Every measurement was performed three times to guarantee accuracy and dependability. Moreover, established protocols assessed the protein content in the tissue homogenate using the Bradford technique.

### 4.18. Efficacy Investigation in In Vitro

#### 4.18.1. Cell Lines and Culturing Conditions

The breast cancer cell lines, namely, the human ER-positive breast adenocarcinoma cells (MCF-7, ATCC^®^ HTB-22™), the drug-resistant cell line MCF-7/ADR (ATCC^®^, HTB-22™), and the normal breast epithelial cells MCF-10A (ATCC^®^ CRL-10317™), were sourced from the American Type Culture Collection (ATCC, Manassas, VA, USA). The cells were cultured in Dulbecco’s Modified Eagle Medium (DMEM) supplemented with 10% fetal bovine serum, 2 mM glutamine, penicillin (100 U/mL), streptomycin (100 μg/mL), and 10 mM HEPES. The cultures were cultured at 37 °C in a 5% CO_2_ atmosphere, with a pH maintained at 7.4. Throughout the experiment, 90% of the medium was replaced every day with fresh, unused media. Subsequently, 100 μL of a cell suspension comprising 1 × 10^5^ cells was introduced into each well of a 96-well microplate for experimental evaluation.

#### 4.18.2. Preparation of WIT and DOX Concentrations

WIT (Ashwagandha root extract, Solgar^®^, Leonia, NJ, USA) and DOX (EBEWE Pharma, Unterach, Austria) were manufactured sterilely, starting with a concentrated solution. The procedure consisted of completely dissolving 10 mg of each specific medication in 100 mL of culture fluid, including 0.5% *v*/*v* DMSO. It is essential to mention that the concentrations of DMSO did not exceed 0.1% (*v*/*v*), a value recognized as safe for the cells. Subsequent concentrations of each medication were created by diluting the first solution with the growth medium. WIT reached 1000, 500, 250, 125, 62.5, and 31.25 μg/mL doses. Concentrations of 100, 50, 25, 12.5, 6.25, and 3.13 μg/mL were found for DOX.

#### 4.18.3. Evaluation of the Toxicological Effects of WIT and DOX Using the MTT Assay

We employed the Methyl Tetrazolium (MTT) assay to assess the harmful effects of WIT and DOX, both individually and in combination, on MCF-7, MCF-7/ADR cells, and MCF-10A. Initially, 100 μL of a cell suspension with a concentration of 1 × 10^5^ cells/mL was dispensed into each well of a 96-well microplate. Subsequently, 100 μL of growth medium containing DMSO (0.1%) was introduced to each well. Thereafter, the samples were subjected to various concentrations of WIT and DOX for 24 h. The control cells were cultivated in a medium with the same DMSO concentration (0.1%) as the experimental cultures for 24 h. Previous investigations have verified that DMSO is not harmful at this dose. Four microplates containing MCF-7, MCF-7/ADR cells, MCF-10A, and growth media were each filled with 100 μL of the tested medications (WIT and DOX) at varying doses. The microplates were then incubated for 24 h at 37 °C, with a CO_2_ concentration of 5% and humidity level of 90%. After the incubation period, 50 μL of MTT stock solution (with a concentration of 5mg/mL in PBS, provided by BIO BASIC CANADA INC., Markham ON, Canada) was introduced into each well and left to incubate for another 4 h. Formazan crystals were solubilized by adding 150 μL of dimethyl sulfoxide (DMSO). The optical density was assessed by measuring absorbance at 560 nm with a Bioline Elisa Microplate Reader (BD-R206, Bioline Technology, Maharashtra, India). The experiment was conducted in triplicate and duplicated thrice. The percentage of viable cells was quantified using the formula (Treated cell absorbance − medium absorbance)/(Control cell absorbance − medium absorbance). The percentage of cell inhibition was determined using the formula (control OD value − experimental OD value)/control OD value × 100. The IC50, indicating the concentration that inhibits 50% of cellular proliferation, was employed to evaluate the drug’s efficacy in suppressing cell growth. Cells cultured without pharmacological agents functioned as the control group. IC50 values were determined utilizing transformed curves (Graph Pad, Prism ISI_ program, Version 9).

#### 4.18.4. Combination Study

Combination tests were performed by subjecting MCF-7/ADR cells to different doses of doxorubicin (DOX), with a constant concentration of either 10 or 20 µg/mL of WIT for 24 h. The DOX concentrations varied from 3.1 to 100 μg/mL. Each experiment was conducted three times with three separate repeats. Cell viability was evaluated, and the combination index (CI) was computed using CompuSyn software (Biosoft, Cambridge, UK), version 1.25. A CI number below 1 signifies synergism, a CI value of 1 signifies additive effects, and a CI value above 1 signifies antagonism in the medication combination.

#### 4.18.5. Annexin-V-FITC Analysis

The Annexin V-FITC Apoptosis Detection Kit (Bio Vision Research Products, 980 Linda Vista Avenue, Mountain View, CA, USA) was employed to assess the capacity of WIT, DOX, and their combination to cause early and late apoptosis in MCF-7/ADR cells, using established protocols [100]. T-75 flasks were inoculated with 1 × 10^6^ cells/mL. MCF-7/ADR cells (1 × 10^6^) were administered WIT (20 μg/mL), DOX (6.2 μg/mL), or a combination of DOX (6.2 μg/mL) and WIT (10 or 20 μg/mL) on 60 mm² culture dishes during the exponential growth phase. The treatments continued for a duration of 24 h. Subsequently, cells were harvested utilizing trypsin and subjected to centrifugation at 1000 rpm for five minutes. The cell pellets were washed with phosphate-buffered saline and resuspended in 400 μL of binding buffer at a density of 1 × 10^6^ cells/mL. Following the addition of 10 μL of propidium iodide and 5 μL of annexin V-FITC, the cells were incubated in darkness at 37 °C for 30 min.

#### 4.18.6. Cell Cycle Analysis

The Becton Dickinson FACS Calibur flow cytometer (BD Biosciences, San Jose, CA, USA) was employed to assess the impact of DOX, WIT, and their combinatorial therapy on the cell cycle distribution of MCF-7/ADR cells [100]. MCF-7/ADR cells (1 × 10^6^) in the exponential growth phase were administered WIT (10 and 20 μg/mL), DOX (6.2 μg/mL), or a combination of both. The control cells received DMSO treatment. The therapy persisted for 48 h. For cell cycle analysis, trypsinized cells (1 × 10^6^ per sample) were harvested, rinsed twice with phosphate-buffered saline, and subsequently fixed in cold 70% ethanol. The cells were subjected to staining with propidium iodide (PI) solution (20 μg/mL PI, 0.1% Triton X-100, and 0.1 μg/mL RNase A) for one hour at 37 °C. Flow cytometry was employed to ascertain cell cycle distribution, and the data were processed utilizing BD Cell Quest Pro software (version 5.1), with 10,000 cells per sample.

### 4.19. Statistical Analysis

The data represent the mean value ± the standard error of the mean (SEM). The statistical studies were performed using SPSS software, version 25.0, developed by IBM, headquartered in Chicago, IL, USA. GraphPad Software is in San Diego, CA, USA. The graphs were generated using Prism 9. The Shapiro–Wilk test was employed to confirm the normality of the data. After doing Tukey’s post hoc analysis to compare the groups, the data were further analyzed using a one-way or two-way analysis of variance (ANOVA). *p*-values less than 0.05 (*p* < 0.05) were deemed statistically significant.

## 5. Conclusions

This study utilized an animal model to show that WIT improved kidney function while lowering oxidative stress, inflammation, and apoptosis of DOX-treated animals. Furthermore, WIT improved the effectiveness of DOX in the treatment of breast cancer by utilizing an in vitro model, specifically targeting drug-resistant breast cancer cells. Therefore, WIT is an effective chemotherapeutic supplement with potent anti-inflammatory, anti-apoptotic, and antioxidant properties (Figure 10). That allows it not only to decrease the possibility of adverse effects but to increase its pharmacological efficacy as well. More research into WIT’s potential as a kidney-protective drug is needed in pharmacodynamic and clinical settings. Moreover, subsequent investigations could utilize sophisticated target-finding techniques, such as protein and PROTAC probes, to further elucidate the mechanisms of action of active compounds. Additional research and clinical trials are required to validate their anticancer properties in humans, which may facilitate their application in cancer chemotherapy, either as monotherapies or in conjunction with other anticancer agents.

## Figures and Tables

**Figure 1 pharmaceuticals-18-00248-f001:**
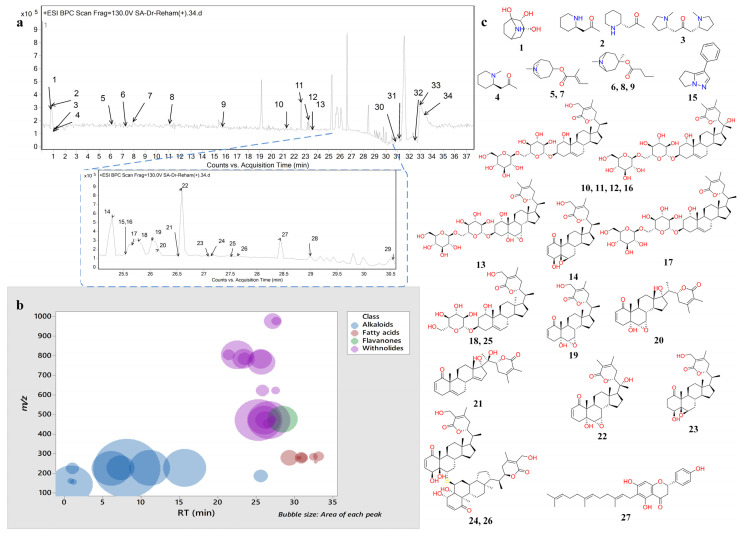
(**a**) The base peak chromatograms (BPCs) of the extract from the roots of WIT in the positive ionization mode. (**b**) A bubble plot shows the observed masses (*m*/*z*) plotted against the retention time (RT), indicating the phytochemical classes and their corresponding relative abundance. (**c**) The structures of the principal metabolites.

**Figure 2 pharmaceuticals-18-00248-f002:**
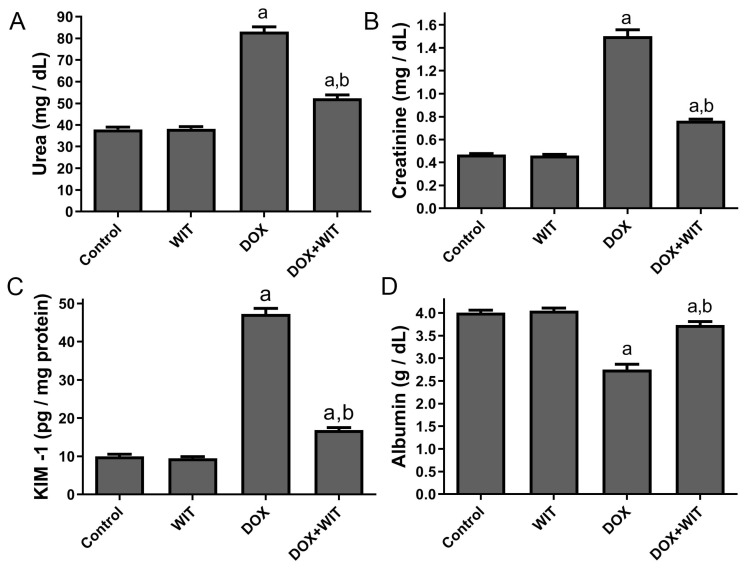
The impact of WIT on the concentrations of urea (**A**), creatinine (**B**), albumin (**D**), and renal level of KIM-1 (**C**) in rats with DOX-induced nephrotoxicity. The results are expressed as the mean ± SEM (n = 6). The significance was assessed using one-way analysis of variance (ANOVA) followed by Tukey’s post hoc test; ^a^ (*p* < 0.05) is statistically significant in comparison to the control group, whereas ^b^ (*p* < 0.05) is statistically significant relative to the DOX group.

**Figure 3 pharmaceuticals-18-00248-f003:**
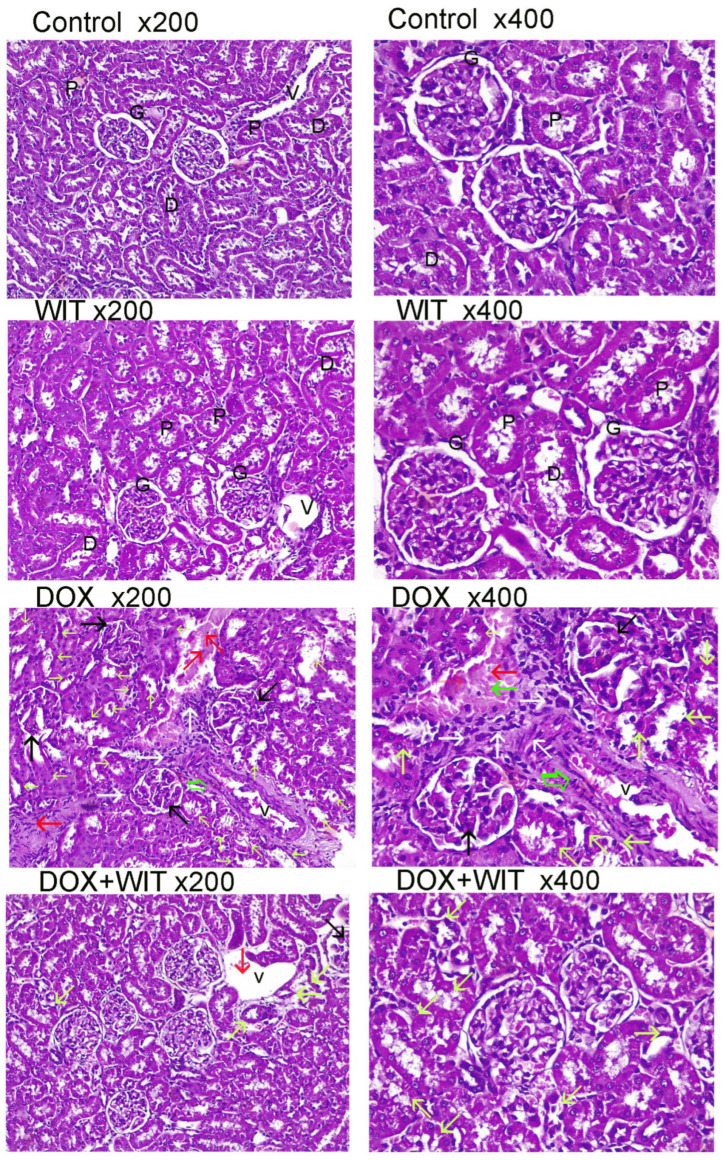
The protective effect of WIT in mitigating kidney damage caused by DOX. The panels display histological alterations in kidney tissue for different groups, seen using H&E staining at a magnification of 200× and 400×. The kidney sections from the WIT group and the control group’s light micrographs display the normal structure of the renal glomerulus corpuscle (G), proximal (P) and distal tubules (D), and blood vessels (V). On the other hand, kidney sections from the DOX groups showed pathological characteristics such as atrophy or distortion of the glomerular capsule (black arrow), widening and degeneration of the tubules (yellow arrow), focal inflammatory cell infiltration between degenerated tubules (white arrow), congestion in blood vessels and focal hemorrhages between tubules (red arrow), and perivascular edema and fibrosis (thick arrow). Kidney section from DOX+WIT showing normal histological structure with mild histopathological changes such as atrophy or distortion of the glomerular capsule (black arrow), tubular dilatation and degenerations (yellow arrows), and congestion and diluted of blood vessels (red arrow).

**Figure 4 pharmaceuticals-18-00248-f004:**
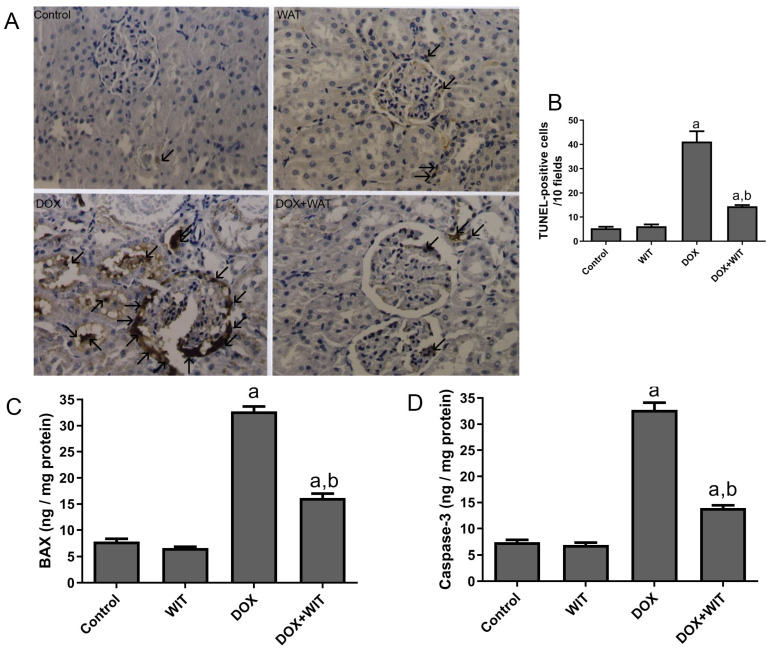
The impact of WIT on cell death (apoptosis) in rats with doxorubicin-induced kidney damage. (**A**) Representative photos depicting immunohistochemical staining with TUNEL-positive cells in kidney sections from all evaluated groups. (**B**) The TUNEL-positive cells in several experimental groups. TUNEL-positive cells were quantified in each segment by counting the number of cells with brown staining (arrows) in 10 areas at 400× magnification. (**C**) The impact of WIT on the renal levels of BAX and (**D**) Caspase-3 in rats with doxorubicin-induced nephrotoxicity. The results are expressed as the mean ± SEM (n = 6). The significance was assessed using one-way analysis of variance (ANOVA) followed by Tukey’s post hoc test; ^a^ (*p* < 0.05) is statistically significant in comparison to the control group, whereas ^b^ (*p* < 0.05) is statistically significant relative to the DOX group.

**Figure 5 pharmaceuticals-18-00248-f005:**
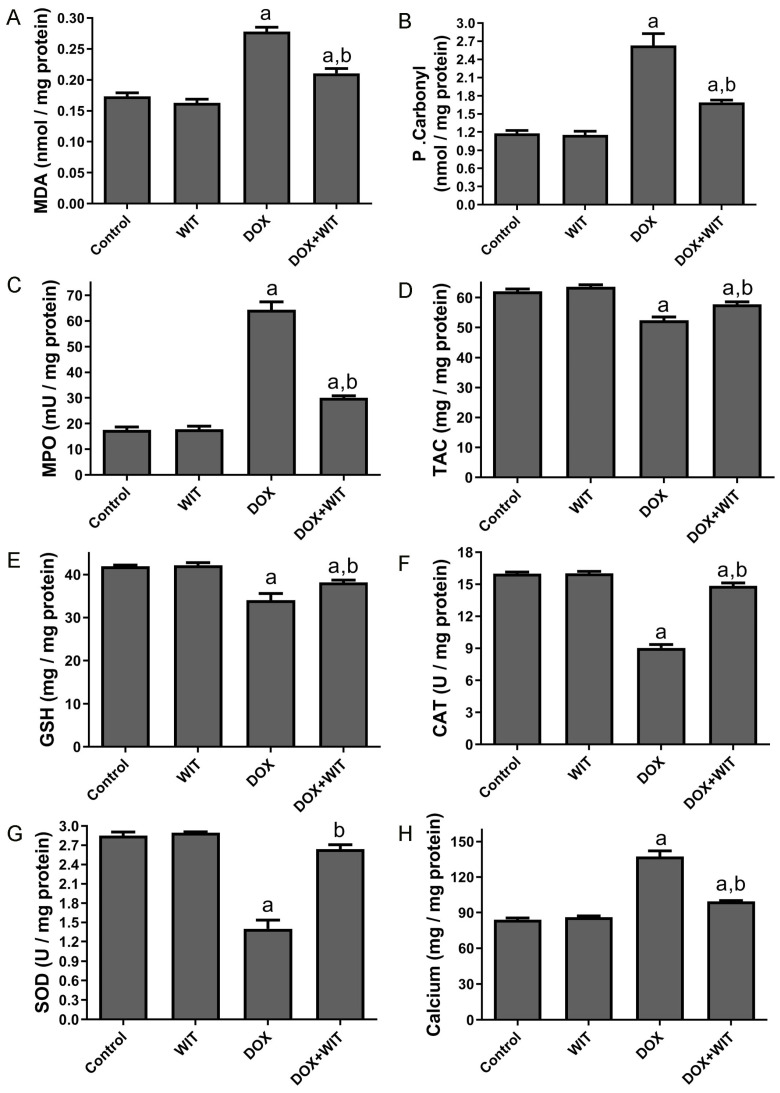
(**A**–**G**) The impact of WIT on oxidative stress markers in the kidneys during DOX-induced nephrotoxicity. (**H**) The impact of WIT on calcium levels in the kidneys during DOX-induced nephrotoxicity. The results are expressed as the mean ± SEM (n = 6). The significance was assessed using one-way analysis of variance (ANOVA) followed by Tukey’s post hoc test; ^a^ (*p* < 0.05) is statistically significant in comparison to the control group, whereas ^b^ (*p* < 0.05) is statistically significant relative to the DOX group.

**Figure 6 pharmaceuticals-18-00248-f006:**
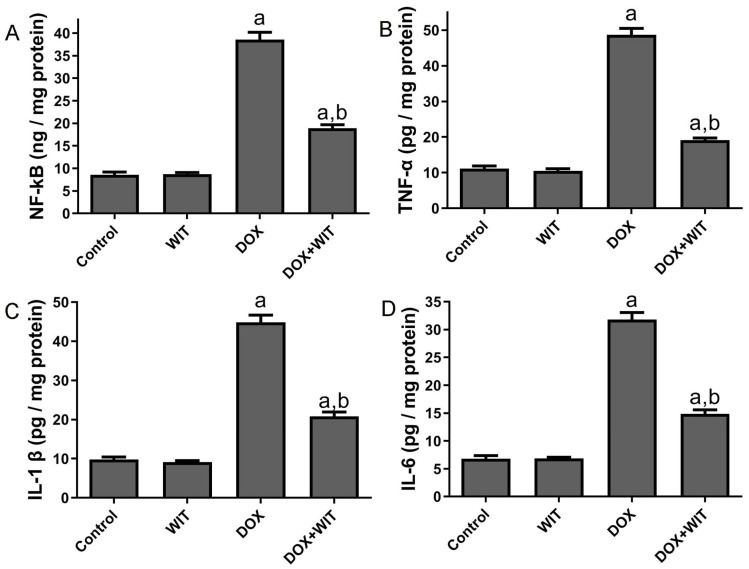
Illustrates the impact of WIT on renal inflammatory markers in rats with DOX-induced nephrotoxicity. The indicators examined include NF-kB (**A**), TNF-α (**B**), IL-1β (**C**), and IL-6 (**D**). The results are expressed as the mean ± SEM (n = 6). The significance was assessed using one-way analysis of variance (ANOVA) followed by Tukey’s post hoc test; ^a^ (*p* < 0.05) is statistically significant in comparison to the control group, whereas ^b^ (*p* < 0.05) is statistically significant relative to the DOX group.

**Figure 7 pharmaceuticals-18-00248-f007:**
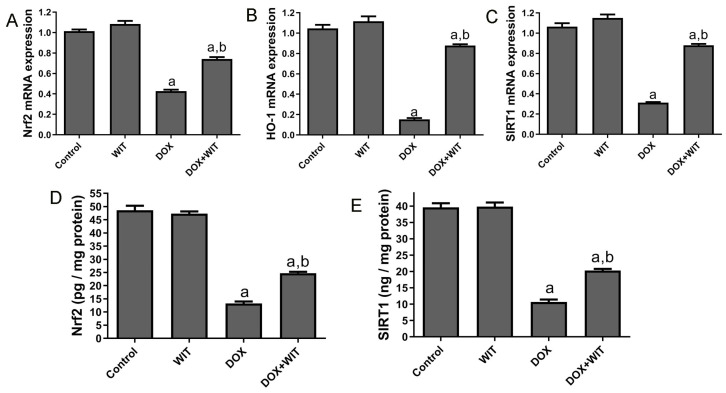
The impact of WIT on the renal expression of Nrf2, HO-1, and SIRT1 in rats treated with DOX. The levels of Nrf2 (**A**), HO-1 (**B**), and SIRT1 (**C**) mRNA in the kidneys of various experimental groups were measured using qRT-PCR and compared to β-actin expression. The renal protein levels of Nrf2 (**D**) and SIRT1 (**E**) were measured using ELISA. The results are expressed as the mean ± SEM (n = 6). The significance was assessed using one-way analysis of variance (ANOVA) followed by Tukey’s post hoc test; ^a^ (*p* < 0.05) is statistically significant in comparison to the control group, whereas ^b^ (*p* < 0.05) is statistically significant relative to the DOX group.

**Figure 8 pharmaceuticals-18-00248-f008:**
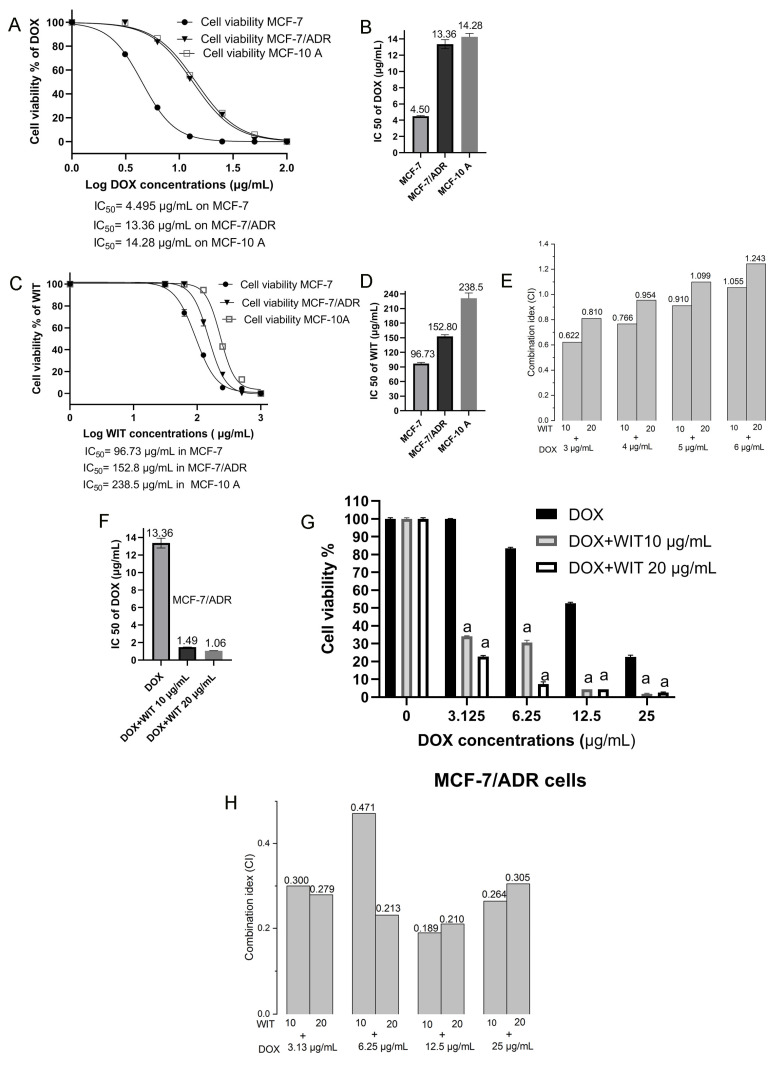
Effects of DOX and WIT on the viability of MCF-7, MCF-7/ADR, and MCF-10A cells. (**A**) MTT experiment demonstrating the effect of DOX on cell viability of MCF-7, MCF-7/ADR, and MCF-10A at various doses 24 h after treatment. (**B**) DOX IC_50_ on MCF-7, MCF-7/ADR, and MCF-10A cells. (**C**) An MTT experiment demonstrated WIT’s cytotoxicity on MCF-7, MCF-7/ADR, and MCF-10A cells after 24 h of treatment. (**D**) The WIT IC_50_ is on MCF-7, MCF-7/ADR, and MCF-10A cells. (**E**) The combination index (CI) values for DOX and WIT in MCF-7/ADR cells were calculated using CompuSyn software (version 1.25). (**F**) The IC_50_ of DOX and DOX-WIT on MCF-7/ADR cells after 24 h of treatment (**G**) The cell viability of DOX and DOX-WIT combination for MCF-7/ADR cells, with ^a^
*p* < 0.05 vs. the DOX treated group. (**H**) CI values were computed using CompuSyn software based on G’s combination data.

**Figure 9 pharmaceuticals-18-00248-f009:**
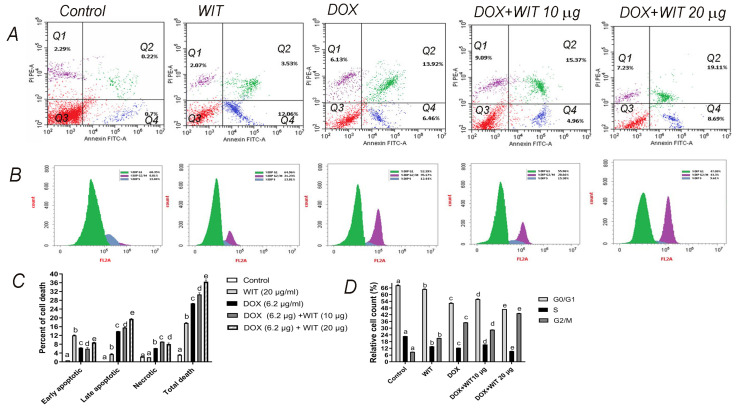
Effect of WIT and DOX combination on apoptosis and cell cycle distribution in MCF-7/ADR cells. (**A**) Flow cytometric study of cells stained with Annexin V and PI to assess apoptosis 24 h post-treatment with 20 μg/mL WIT, 6.2 μg/mL DOX, and combinations of 10 and 20 μg/mL WIT with 6.2 μg/mL DOX. The percentage of cells in each quadrant corresponds to a particular cell type: necrotic cells (Q1), live cells (Q3), early apoptotic cells (Q4), and late apoptotic cells (Q2). (**B**) Flow cytometric examination shows the distribution of cells in the G0/G1, S, and G2/M phases following treatment with DMSO (less than 1% as a negative control), WIT, DOX, and DOX+WIT at concentrations of 10 and 20 μg/mL. (**C**) Calculating the rates of apoptosis and necrosis with the calibration from (**A**). Values are expressed as a proportion of total cells. Each point shows the mean ± SEM (n = 3). (**D**) Calculating the percentage of cell dispersion across different cell cycle phases. Each bar represents the mean ± SEM of three independent studies. A two-way ANOVA and Tukey’s multiple comparison tests evaluated significant differences among experimental groups. Non-identical letters signify a statistically significant difference between experimental groups (*p* < 0.05).

**Figure 10 pharmaceuticals-18-00248-f010:**
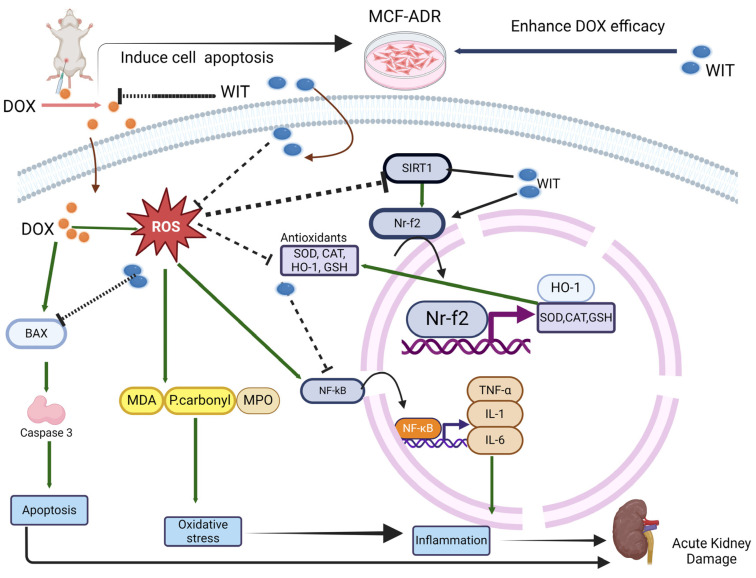
Schematic diagram of the antioxidant, anti-inflammatory, and antiapoptotic effects of WIT against DOX-induced nephrotoxicity and the enhancement of the efficacy of DOX in treating breast cancer. CAT, catalase; BAX, Bcl-2–associated X protein; DOX, doxorubicin; GSH, reduced glutathione; HO1, haem oxygenase-1; IL-1β, interleukin-1 beta; IL-6, interleukin-6; KIM-1, kidney injury molecule-1; MPO, myeloperoxidase; erythroid-2-related factor 2; P. carbonyl, Portion carbonyl; SIRT-1, Sirtuin-1; SOD, superoxide dismutase; TAC, total antioxidant capacity; MDA, malondialdehyde; TNF-α, tumor necrosis factor-α; WIT, *Withania somnifera*.

**Table 1 pharmaceuticals-18-00248-t001:** Metabolites characterized in *WIT* root extract.

Peak No.	RT (min)	[M+H]^+^	M	Molecular Formula	Score	Error (ppm)	Error (mDa)	Main Fragments	DBE	Proposed Compound	Class	Area	%
1	0.8	160.0968	159.0900	C_7_H_13_NO_3_	80.6	−2.6	−0.4	N.D.	2	Calystegin A3	A	4.50 ×10^4^	0.10
2	0.8	142.1224	141.1154	C_8_H_15_NO	86.8	−1.6	−0.2	124.1114, 98.0966, 84.0811	2	Pelletierine/Isopelletierine	A	3.39 × 10^6^	7.91
3	1.0	225.1964	224.1891	C_13_H_24_N_2_O	89.2	−1.1	−0.2	N.D.	3	Cuscohygrine	A	2.57 × 10^5^	0.60
4	1.1	156.1384	155.1309	C_9_H_17_NO	78.9	0.89	0.14	N.D.	2	N-Methylpelletierine	A	3.73 × 10^4^	0.09
5	6.1	224.1645	223.1574	C_13_H_21_NO_2_	98	−0.7	−0.2	124.1120, 96.0809, 95.0844	4	Tigloidine isomer I	A	2.85 × 10^6^	6.65
6	7.3	226.1807	225.1728	C_13_H_23_NO_2_	97.8	−1.8	−0.4	124.1125, 96.0806	3	3-Methylbutyryloxytropane isomer I	A	1.45 × 10^6^	3.37
7	8.1	224.1646	223.1574	C_13_H_21_NO_2_	98.4	−1.2	−0.3	124.1120, 96.0810, 95.0849, 93.0702	4	Tigloidine Isomer II	A	8.23 × 10^6^	19.18
8	11.0	226.1801	225.1728	C_13_H_23_NO_2_	99.6	0.17	0.04	124.1122, 96.0818, 95.0856, 93.0701	3	3-Methylbutyryloxytropane isomer II	A	2.95 × 10^6^	6.87
9	15.7	226.1802	225.1728	C_13_H_23_NO_2_	98.5	0.28	0.06	124.1107, 95.0845, 93.0682	3	3-Methylbutyryloxytropane isomer III	A	3.37 × 10^6^	7.86
10	21.3	805.3973 *	782.4089	C_40_H_62_O_15_	97.8	0.66	0.53	N.D.	13	Withanoside IV/VI isomer I	W	2.17 × 10^5^	0.51
11	22.5	805.3978 *	782.4089	C_40_H_62_O_15_	90.3	3.36	2.7	643.3412, 457.2262	13	Withanoside IV/VI isomer II	W	1.89 × 10^6^	4.40
12	23.3	805.3996 *	782.4089	C_40_H_62_O_15_	96.9	1.62	1.3	787.3441, 643.3557, 457.2262	13	Withanoside IV/VI isomer III	W	8.78 × 10^5^	2.05
13	23.5	821.3933 *	798.4038	C_40_H_62_O_16_	85.3	4.17	3.42	785.3759, 685.7805	13	Withanoside II	W	3.29 × 10^5^	0.77
14	25.3	493.2567 *	470.2679	C_28_H_38_O_6_	98.4	−1.3	−0.6	452.1819, 281.1534	10	Withaferin A	W	4.16 × 10^6^	9.68
15	25.5	185.1076	184.0426	C_12_H_12_N_2_	96.9	−2	−0.4	N.D.	8	Withasomnine	A	3.22 × 10^5^	0.75
16	25.5	805.3996 *	782.4089	C_40_H_62_O_15_	96.9	1.62	1.3	787.3441, 643.3557, 457.2262	13	Withanoside IV/VI isomer IV	W	8.78 × 10^5^	2.05
17	25.7	789.4033 *	766.414	C_40_H_62_O_14_	93.8	−1.9	−1.4	627.3371, 465.2947	10	Withanoside V	W	1.37 × 10^6^	3.19
18	25.8	621.3646	620.3568	C_34_H_52_O_10_	91.8	−1.2	−0.8	N.D.	9	Physagulin D isomer I	W	2.87 × 10^6^	0.67
19	26.0	493.2579 *	470.2679	C_28_H_38_O_6_	98.4	−1.4	−0.6	431.2489, 340.1996, 263.1527	10	12-Deoxywithastramonolide	W	1.85 × 10^6^	4.31
20	26.2	493.2571 *	470.2679	C_28_H_38_O_6_	97.5	0.22	0.11	453.1824, 434.8953, 471.0239, 398.8221, 380.8305	10	Withanone	W	7.09 × 10^5^	1.65
21	26.6	453.2639	452.2574	C_28_H_36_O_5_	78.7	−2.6	−1.7	N.D.	11	Withanolide L	W	2.01 × 10^5^	0.47
22	26.6	493.2571 *	470.2679	C_28_H_38_O_6_	90.8	−0.4	−1.1	435.0745, 263.1435	10	Withanolide A	W	3.39 × 10^6^	7.90
23	27.1	495.2693 *	472.2814	C_28_H_40_O_6_	84.5	0.42	0.21	N.D.	9	2,3-Dihydrowithaferin A	W	1.22 × 10^5^	0.28
24	27.1	997.5120 *	974.5221	C_56_H_78_O_12_S	90.9	−0.7	−0.7	979.4905, 935.5011, 468.1338	18	Ashwagandhanolide isomer I	W	4.95 × 10^5^	1.15
25	27.5	621.3649	620.3568	C_34_H_52_O_10_	80.9	−1.8	−1.1	N.D.	9	Physagulin D isomer II	W	1.19 × 10^5^	0.28
26	27.6	997.5102 *	974.5221	C_56_H_78_O_12_S	78	0.55	0.54	N.D.	18	Ashwagandhanolide isomer II	W	1.31 × 10^5^	0.31
27	28.4	477.2605	476.2546	C_30_H_36_O_5_	90.5	3.42	1.63	271.06	13	Farnesylnaringenin	Fl	1.69 × 10^6^	3.95
28	29.3	279.2324	278.2252	C_18_H_30_O_2_	98.5	−2.1	−0.6	209.1509	4	Linolenic acid isomer I	Fa	5.40 × 10^5^	1.26
29	30.7	279.2313	278.2252	C_18_H_32_O_2_	82.7	−6.6	−1.9	N.D.	3	Linoleic acid	Fa	2.04 × 10^4^	0.05
30	30.8	279.2313	278.2252	C_18_H_30_O_2_	74.8	−2.5	−0.7	261.2221, 243.2131, 209.1542	4	Linolenic acid isomer II	Fa	2.90 × 10^5^	0.68
31	31.0	279.2317	278.2252	C_18_H_30_O_2_	82.7	−0.5	−0.1	N.D.	4	Linolenic acid isomer III	Fa	1.69 × 10^5^	0.39
32	32.5	283.2645	282.2576	C_18_H_34_O_2_	81.8	−6.2	−1.7	N.D.	2	Oleic acid	Fa	7.82 × 10^4^	0.18
33	32.7	257.2485	256.2402	C_16_H_32_O_2_	96.1	−3.9	−1	236.9892	1	Palmitic acid	Fa	1.87 × 10^4^	0.04
34	33.1	285.2780	284.2708	C_18_H_36_O_2_	85.6	2.55	0.73	N.D.	1	Octadecanoic acid	Fa	1.74 × 10^5^	0.41

A, Alkaloids; Fa, Fatty acids; Fl, Flavanones; W, Withanolides; DBE, Double bond equivalence; * ion with sodium adduct; N.D., undetected. Peak area: lowest value 
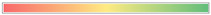
 highest value.

**Table 2 pharmaceuticals-18-00248-t002:** The impact of treatment with DOX and WIT on body weight, kidney weight, and relative kidney weight.

Groups	Body Weight (g)		Kidney Weight (g)	Relative Kidney Weight
	Final	Body Gain		
Control	266.3 ± 7.34	46.13 ± 4.55	2.13 ± 0.056	0.807 ± 0.026
WIT	254.3 ± 12.61	63.33 ± 5.54 ^b^	1.98 ± 0.098	0.795 ± 0.024
DOX	146.0 ± 3.51 ^a^	−34.67 ± 3.90 ^a^	0.93 ± 0.021 ^a^	0.641 ± 0.023 ^a^
DOX + WIT	169.7 ± 5.49 ^a,b^	−20.50 ± 1.30 ^a^	1.31 ± 0.063 ^a,b^	0.766 ± 0.026 ^b^

The results are expressed as the mean ± SEM (n = 6). The significance was assessed using one-way analysis of variance (ANOVA) followed by Tukey’s post hoc test.; ^a^ (*p* < 0.05) is statistically significant in comparison to the control group, whereas ^b^ (*p* < 0.05) is statistically significant relative to the DOX group.

**Table 3 pharmaceuticals-18-00248-t003:** Histopathological damage scores of the DOX and WIT in kidney tissues.

Histopathological Changes	Control and WIT Group	DOX Group	DOX + WIT Group
Distortion of the glomerular capsule	0	1.0 ± 0.0 ^a^	0.6 ± 0.40
Atrophy of the glomerular capsule	0	2.4 ± 0.25 ^a^	1.2 ± 0.24 ^a^
Tubular dilatation and protein casts	0	1.8 ± 0.20 ^a^	0.6 ± 0.25 ^b^
Tubular degeneration	0	2.8 ± 0.20 ^a^	0.6 ± 0.24 ^b^
Congestion in blood vessels and interstitial hemorrhage	0	2.6 ± 0.25 ^a^	1.2 ± 0.20 ^a^
Focal inflammation	0	2.6 ± 0.25 ^a^	0.6 ± 0.20 ^b^
Perivascular edema and fibrosis	0	2.6 ± 0.25 ^a^	0.6 ± 0.24 ^b^

The results are expressed as the mean ± SEM (n = 5). Significance was assessed using the non-parametric Kruskal Wallis H test, followed by post hoc pairwise comparisons: ^a^ (*p* < 0.05) is statistically significant in comparison to the control group, whereas ^b^ (*p* < 0.05) is statistically significant relative to the DOX group.

## Data Availability

The publication contains all of the data used in this study. If you have any further questions, you can contact the appropriate authors.

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
