# Peer review of "Withania somnifera Ameliorates Doxorubicin-Induced Nephrotoxicity and Potentiates Its Therapeutic Efficacy Targeting SIRT1/Nrf2, Oxidative Stress, Inflammation, and Apoptosis"

_pharmaceuticals, 2025, doi:10.3390/ph18020248_

Round 1
Reviewer 1 Report
Comments and Suggestions for Authors
The manuscript explores the potential protective benefits of a standardized WIT extract against DOX-induced renal damage in vivo and examines the synergistic effects of combining WIT with DOX to enhance therapeutic efficacy in breast cancer cells (MCF7-ADR). While the study is intriguing and addresses an important area of research, several aspects require further refinement and clarification to meet publication standards.
The manuscript requires careful revision to address the following points:
-
Ensure that the terms in vitro and in vivo are consistently written in italics throughout the text for proper scientific formatting.
-
The Materials and Methods section lacks details on how the plant sample was prepared for the identification of phytoconstituents. Please include a comprehensive description of the preparation process to enhance the reproducibility of the study.
-
In Figure 4C and 4D, the value obtained for WIT is lower than that of the control. Please provide a thorough explanation and justification for this observation to clarify its biological or experimental relevance.
-
The Conclusions section needs significant improvement. It should not only summarize the findings but also address the limitations of the study and provide perspectives for future research. This will offer a more balanced and forward-looking conclusion to the manuscript.
Author Response
Comment 1
Ensure that the terms in vitro and in vivo are consistently written in italics throughout the text for proper scientific formatting
Response:
As recommended by the reviewer, this has been modified. We have also ensured that all in vitro and in vivo terms are in italics throughout the text. Thank you very much.
Comment 2
The Materials and Methods section lacks details on how the plant sample was prepared for the identification of phytoconstituents. Please include a comprehensive description of the preparation process to enhance
the reproducibility of the study.
Response:
Thank you very much for this constructive advice. Based on your advice, a method for preparing the plant sample to identify its components has been added. A comprehensive description of the preparation process has also been included.
Comment 3
In Figures 4C and 4D, the value obtained for WIT is lower than that of the control. Please provide a thorough explanation and justification for this observation to clarify its biological or experimental relevance.
Response:
Figures 4C and 4D illustrate the renal levels of BAX and (D) Caspase-3 in rats from the four groups. Upon analyzing the data using SPSS software, there was no significant difference between the values of the rat group that received WIT and the control group. I am enclosing here the analysis data. The statistician has two tables for analysis: mean value ± the standard error of the mean (SEM). The other table is for Tukey's post-hoc analysis to compare the groups, and consequently, there is no justification for commenting on that section.
SPSS Report Mean value ± the standard error of the mean (SEM |
|||
groups |
BAX |
Caspase |
|
DOX |
Mean |
32.7500 |
32.7167 |
N |
6 |
6 |
|
Std. Error of Mean |
.93265 |
1.38814 |
|
DOX+WIT |
Mean |
16.1333 |
13.9167 |
N |
6 |
6 |
|
Std. Error of Mean |
.84997 |
.53004 |
|
WIT |
Mean |
6.5667 |
6.8833 |
N |
6 |
6 |
|
Std. error of Mean |
.25517 |
.41426 |
|
Control |
Mean |
7.8000 |
7.3833 |
N |
6 |
6 |
|
Std. Error of Mean |
.53166 |
.44528 |
|
Total |
Mean |
15.8125 |
15.2250 |
N |
24 |
24 |
|
Std. Error of Mean |
2.20269 |
2.21564 |
Multiple Comparisons |
|||||||
Tukey's post-hoc analysis |
|||||||
Dependent Variable |
(I) groups |
(J) groups |
Mean Difference (I-J) |
Std. Error |
Sig. |
95% Confidence Interval |
|
Lower Bound |
Upper Bound |
||||||
BAX |
DOX |
2.000 |
16.61667* |
.98490 |
.000 |
13.8600 |
19.3733 |
3.000 |
26.18333* |
.98490 |
.000 |
23.4267 |
28.9400 |
||
4.000 |
24.95000* |
.98490 |
.000 |
22.1933 |
27.7067 |
||
DOX+WIT |
1.000 |
-16.61667-* |
.98490 |
.000 |
-19.3733- |
-13.8600- |
|
3.000 |
9.56667* |
.98490 |
.000 |
6.8100 |
12.3233 |
||
4.000 |
8.33333* |
.98490 |
.000 |
5.5767 |
11.0900 |
||
WIT |
1.000 |
-26.18333-* |
.98490 |
.000 |
-28.9400- |
-23.4267- |
|
2.000 |
-9.56667-* |
.98490 |
.000 |
-12.3233- |
-6.8100- |
||
4.000 |
-1.23333- |
.98490 |
.602 |
-3.9900- |
1.5233 |
||
Control |
1.000 |
-24.95000-* |
.98490 |
.000 |
-27.7067- |
-22.1933- |
|
2.000 |
-8.33333-* |
.98490 |
.000 |
-11.0900- |
-5.5767- |
||
3.000 |
1.23333 |
.98490 |
.602 |
-1.5233- |
3.9900 |
||
Caspase |
DOX |
2.000 |
18.80000* |
1.13529 |
.000 |
15.6224 |
21.9776 |
3.000 |
25.83333* |
1.13529 |
.000 |
22.6557 |
29.0109 |
||
4.000 |
25.33333* |
1.13529 |
.000 |
22.1557 |
28.5109 |
||
DOX+WIT |
1.000 |
-18.80000-* |
1.13529 |
.000 |
-21.9776- |
-15.6224- |
|
3.000 |
7.03333* |
1.13529 |
.000 |
3.8557 |
10.2109 |
||
4.000 |
6.53333* |
1.13529 |
.000 |
3.3557 |
9.7109 |
||
WIT |
1.000 |
-25.83333-* |
1.13529 |
.000 |
-29.0109- |
-22.6557- |
|
2.000 |
-7.03333-* |
1.13529 |
.000 |
-10.2109- |
-3.8557- |
||
4.000 |
-.50000- |
1.13529 |
.971 |
-3.6776- |
2.6776 |
||
Control |
1.000 |
-25.33333-* |
1.13529 |
.000 |
-28.5109- |
-22.1557- |
|
2.000 |
-6.53333-* |
1.13529 |
.000 |
-9.7109- |
-3.3557- |
||
3.000 |
.50000 |
1.13529 |
.971 |
-2.6776- |
3.6776 |
||
*. The mean difference is significant at the 0.05 level. |
Comment 5
The Conclusions section needs significant improvement. It should not only summarize the findings, but also address the limitations of the study and provide perspectives for future research. This will offer a more balanced and forward-looking conclusion to the manuscript
Response:
Thank you very much. Based on your constructive advice, the conclusions section has been improved. The study's limitations have been addressed and presented, as well as perspectives for future research. The following has been added and highlighted in yellow. More research into WIT's potential as a kidney-protective drug is needed, both in pharmacodynamic and clinical settings. Moreover, subsequent investigations could utilize sophisticated target-finding techniques, such as protein and PROTAC probes, to further elucidate the mechanisms of action of active compounds. Additional research and clinical trials are required to validate their anticancer properties in humans, which may facilitate their application in cancer chemotherapy, either as monotherapies or in combination with other anticancer agents.
Comment 6
The English could be improved to more clearly express the research.
Response:
The language has been modified and improved.
Reviewer 2 Report
Comments and Suggestions for Authors
This paper investigates the effects of Withania somnifera (WIT) on doxorubicin (DOX)-induced nephrotoxicity and its combination with DOX in breast cancer cells. In an animal model, WIT reduced kidney damage markers, oxidative stress, inflammation, and apoptosis in DOX-treated rats. Chemical analysis of WIT root extract identified 34 compounds. In vitro, WIT and DOX showed synergistic cytotoxicity against MCF-7/ADR cells, increasing apoptosis and cell cycle arrest. WIT's antioxidant, anti-inflammatory, and anti-apoptotic properties are likely due to its bioactive compounds. Overall, WIT shows potential as a chemotherapeutic supplement to enhance DOX's efficacy and reduce toxicity. I recommend that the journal accept the manuscript after the authors make the following revisions:
1. Formatting Issues: On pages 4.11 and 4.18, the line spacing in some sections is noticeably inconsistent with other paragraphs. Additionally, on page 4.18, under "2. Preparation of WIT and DOX Concentrations," there is an unnecessary indentation.
2.Attention to Subscripts: Ensure that subscripts are correctly used, such as in CO2, H2O2, and IC50, where the numbers should be subscripted.
3. Table Formatting: Use a three-line table format for all tables.
4. Figure 10: Add explanations for the symbols used in Figure 10.
5. Figure 8.E: The word "cells" is repeated in the annotation of Figure 8.E.
6. Mechanism Study and Future Directions**: While the components of WIT have been preliminarily identified, the study does not explore the mechanisms related to these components. The authors should at least discuss this in the outlook section. Furthermore, future research could employ advanced target discovery technologies, such as proteomics and PROTAC probes, to further investigate the mechanisms of action of the active compounds. It is recommended to add this discussion and cite the relevant literature (European Journal of Medicinal Chemistry, 276, 2024, 116725. doi: 10.1016/j.ejmech.2024.116725).
Comments on the Quality of English LanguageThe English could be improved to more clearly express the research.
Author Response
Reviewer # 2
Comment 1
Formatting Issues: On pages 4.11 and 4.18, the line spacing in some sections is noticeably inconsistent with other paragraphs. Additionally, on page 4.18, under "2. Preparation of WIT and DOX Concentrations," there is an unnecessary indentation.
Response:
We apologize for the formatting problems. This problem has now been corrected throughout the manuscript, and the line spacing in all sections has been corrected to be consistent with other paragraphs. In addition, unnecessary indentation has been removed.
Comment 2
Attention to Subscripts: Ensure that subscripts are correctly used, such as in CO2, H2O2, and IC50, where the numbers should be subscripted.
Response:
We apologize for this typographical error. As pointed out by the reviewer, subscripts have been corrected throughout the manuscript.
Comment 3
Table Formatting: Use a three-line table format for all tables.
Response:
A three-line table format is now used for all tables.
Comment 4
Add explanations for the symbols used in Figure 10.
Response:
At the request of the reviewer, an explanation of the symbols used in Figure 10 has now been added.
Comment 5
Figure 8.E: The word "cells" is repeated in the annotation of Figure 8.E.
Response:
We apologize for this error. The duplicate word has been deleted. It has been fixed to be on cell viability.
Comment 6
Mechanism Study and Future Directions**: While the components of WIT have been preliminary identified, the study does not explore the mechanisms related to these components. The authors should at least discuss this in the outlook section. Furthermore, future research could employ advanced target discovery technologies, such as proteomics and PROTAC probes, to further investigate the mechanisms of action of the active compounds. It is recommended to add this discussion and citing the relevant literature (European Journal of Medicinal Chemistry, 276, 2024, 116725. doi: 10.1016/j.ejmech.2024.116725).
Response:
Thank you very much for the valuable advice. We have pointed out the importance of using new techniques such as proteomics and PROTAC probes, to further investigate the mechanisms of action of the active compounds, and we cited the wonderful reference that suggested.
At the end of the discussion, the following was added: The encouraging outcomes of WIT with active constituents and their efficacy in enhancing DOX's effectiveness against cancer cells necessitate the implementation of a novel drug development strategy known as "Proteolysis Targeting Chimera" (PROTAC) (Yan et al., 2024). Currently, reported PROTACs for target discovery predominantly emphasize natural products and pharmaceuticals. The utilization of PROTAC technology in drug target discovery represents a significant research avenue, particularly in the identification of targets for natural products and novel drugs. A new multi-target identification approach termed “Degradation-based Protein Profiling” (DBPP) may facilitate the identification of targets for these diverse active compounds in WIT and the development of promising agents for cancer treatment, inflammation targeting, and other beneficial outcomes.
Reference
Yan, S.; Zhang, G.; Luo, W.; Xu, M.; Peng, R.; Du, Z.; Liu, Y.; Bai, Z.; Xiao, X. and Qin, S. PROTAC technology: From drug development to probe technology for target deconvolution. Eur. J. Med. Chem., 2024, 276, 116725.
Comment 6
The English could be improved to more clearly express the research
The manuscript has been revised thoroughly to improve the English langua
Round 2
Reviewer 1 Report
Comments and Suggestions for Authors
Agree with revised manuscript